# LayerNAS: Neural Architecture Search in Polynomial Complexity

## Abstract

Neural Architecture Search (NAS) has become a popular method for discovering effective model architectures, especially for target hardware. As such, NAS methods that find optimal architectures under constraints are essential. In our paper, we propose LayerNAS to address the challenge of multi-objective NAS by transforming it into a combinatorial optimization problem, which effectively constrains the search complexity to be polynomial. LayerNAS rigorously derives its method from the fundamental assumption that modifications to previous layers have no impact on the subsequent layers. When dealing with search spaces containing $L$ layers that meet this requirement, the method performs layerwise-search for each layer, selecting from a set of search options $\mathbb{S}$. LayerNAS groups model candidates based on one objective, such as model size or latency, and searches for the optimal model based on another objective, thereby splitting the cost and reward elements of the search. This approach limits the search complexity to $O(H \cdot |\mathbb{S}| \cdot L)$, where $H$ is a constant set in LayerNAS. Our experiments show that LayerNAS is able to consistently discover superior models across a variety of search spaces in comparison to strong baselines, including search spaces derived from NATS-Bench, MobileNetV2 and MobileNetV3.

## 1 Introduction

With the surge of ever-growing neural models used across all ML-based disciplines, the efficiency of neural networks is becoming a fundamental factor in their success and applicability. A carefully crafted architecture can achieve good quality while maintaining efficiency during inference. However, designing optimized architectures is a complex and time-consuming process – this is especially true when multiple objectives are involved, including the model's performance and one or more cost factors reflecting the model's size, Multiply-Adds (MAdds) and inference latency. Neural Architecture Search (NAS), is a highly effective paradigm for dealing with such complexities. NAS automates the task and discovers more intricate and complex architectures than those that can be found by humans. Additionally, recent literature shows that NAS allows to search for optimal models under specific constraints (e.g., latency), with remarkable applications on architectures such as MobileNetV3 (Howard et al., 2019), EfficientNet (Tan & Le, 2019) and FBNet (Wu et al., 2019).

Most NAS algorithms encode model architectures using a list of integers, where each integer represents a selected search option for the corresponding layer. In particular, notice that for a given model with $L$ layers, where each layer is selected from a set of search options $\mathbb{S}$, the search space contains $O(|\mathbb{S}|^L)$ candidates with different architectures. This exponential complexity presents a significant efficiency challenge for NAS algorithms.

In this paper, we present LayerNAS, an algorithm that addresses the problem of Neural Architecture Search (NAS) through the framework of combinatorial optimization. The proposed approach decouples the constraints of the model and the evaluation of its quality, and explores the factorized search space more effectively in a layerwise manner, reducing the search complexity from exponential to polynomial.

LayerNAS rigorously derives the method from the fundamental assumption: high-performing models when searching for $\text{layer}_i$ can be constructed from one of the models in $\text{layer}_{i-1}$. For search spaces that satisfy this assumption, LayerNAS enforces a directional search process from the first layer to

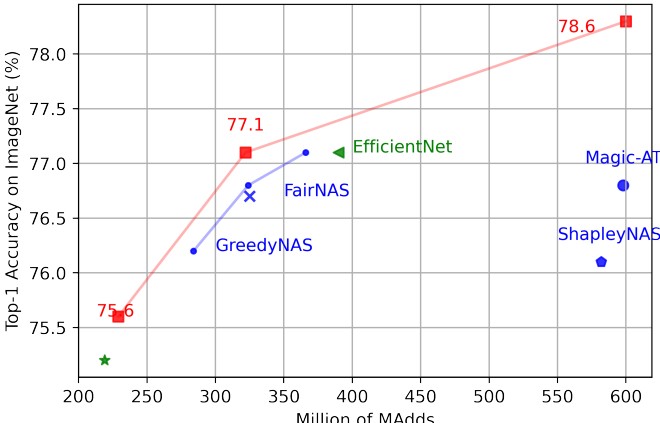

Figure 1: Comparison with baseline models and NAS methods.

the last layer. The directional layerwise search makes the search complexity $O(C \cdot |\mathbb{S}| \cdot L)$, where $C$ is the number of candidates to search per layer.

For multi-objective NAS problems, LayerNAS treats model constraints and model quality as separate metrics. Rather than utilizing a single objective function that combines multi-objectives, LayerNAS stores model candidates by their constraint metric value. Let $\mathcal{M}_{i,h}$ be the best model candidate for $\text{layer}_i$ with $\text{cost} = h$. LayerNAS searches for high-performing models under different constraints in the next layer by adding the cost of the selected search option for next layer to the current layer, i.e., $\mathcal{M}_{i,h}$. This transforms the problem into the following combinatorial optimization problem: *for a model with $L$ layers, what is the optimal combination of options for all layers needed to achieve the best quality under the cost constraint?* If we bucketize the potential model candidates by their cost, the search space is limited to $O(H \cdot |\mathbb{S}| \cdot L)$, where $H$ is number of buckets per layer. In practice, capping the search at 100 buckets achieves reasonable performance. Since this holds $H$ constant, it makes the search complexity polynomial.

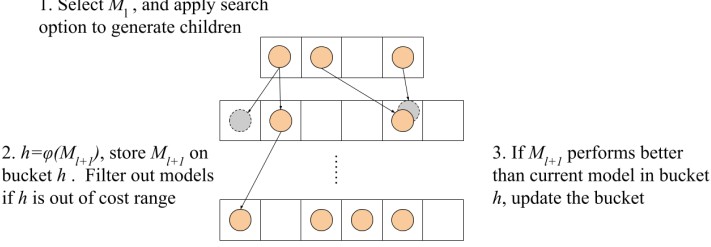

Figure 2: Illustration of the LayerNAS Algorithm described in Algorithm 1. For each layer: (1) select a model candidate from current layer and generate children candidates; (2) store the candidate in corresponding bucket, and filter out candidates not in the target objective range; (3) update the model in the bucket if there's a candidate with better quality; and finally, move to the next layer.

Our contributions can be summarized as follows:

- We propose LayerNAS, an algorithm that transforms the multi-objective NAS problem to a combinatorial optimization problem. This is a novel formulation of NAS.

- LayerNAS is directly designed to tackle the search complexity of NAS, and reduce the search complexity from $O(|\mathbb{S}|^L)$ to $O(H \cdot |\mathbb{S}| \cdot L)$, where $H$ is a constant defined in the algorithm.

- We demonstrate the effectiveness of LayerNAS by identifying high-performing model architectures under various Multiply-Adds (MAdds) constraints, by searching through search spaces derived from MobileNetV2 (Sandler et al., 2018) and MobileNetV3 (Howard et al., 2019).

## 2 RELATED WORK

The survey by Elsken et al. (2019) categorizes methods for Neural Architecture Search into three dimensions: search space, search strategy, and performance estimation strategy. The formulation of NAS as different problems has led to the development of a diverse array of search algorithms. Bayesian Optimization is first adopted for hyper-parameter tuning (Bergstra et al., 2013; Domhan et al., 2015; Falkner et al., 2018; Kandasamy et al., 2018). Reinforcement Learning is utilized for training an agent to interact with a search space (Zoph & Le, 2017; Pham et al., 2018; Zoph et al., 2018; Jaafra et al., 2019). Evolutionary algorithms (Liu et al., 2021; Real et al., 2019) have been employed by encoding model architectures to DNA and evolving the candidate pool. ProgressiveNAS (Liu et al., 2018a) uses heuristic search to gradually build models by starting from simple and shallow model architectures and incrementally adding more operations to arrive at deep and complex final architectures. This is in contrast to LayerNAS, which iterates over changes in the layers of a full complex model.

Recent advancements in mobile image models, such as MobileNetV3 (Howard et al., 2019), EfficientNet (Tan & Le, 2019), FBNet (Wu et al., 2019), are optimized by NAS. The search for these models is often constrained by metrics such as FLOPs, model size, latency, and others. To solve this multi-objective problem, most NAS algorithms (Tan et al., 2019; Cai et al., 2018) design an objective function that combines these metrics into a single objective. LEMONADE (Elsken et al., 2018) proposes a method to split two metrics, and searches for a Pareto front of a family of models. Once-for-All (Cai et al., 2020) proposes progressive shrinking algorithm to efficiently find optimal model architectures under different constraints.

Larger models tend to have better performance compared to smaller models. However, the increased size of models also means increased computational resource requirement. As a result, the optimization of neural architectures within constrained resources is an important and meaningful aspect of NAS problems, which can be solved as multi-objective optimization (Hsu et al., 2018). There is increasing interest in treating NAS as a compression problem (Zhou et al., 2019; Yu & Huang, 2019) from an over-sized model. These works indicate that compressing with different configurations on each layer leads to a model better than uniform compression. Here, NAS can be used to search for optimal configurations (He et al., 2018; Liu et al., 2019; Wang et al., 2019).

The applicability of NAS is significantly influenced by the efficiency of its search process. One-shot algorithms (Liu et al., 2018b; Cai et al., 2018; Bender et al., 2018; 2020) provide a novel approach by constructing a supernet from the search space to perform more efficient NAS. However, this approach has limit on number of branches in supernet due to the constraints of supernet size. The search cost is not only bounded by the complexity of search space, but also the cost of training under "train-and-eval" paradigm. Training-free NAS (Mellor et al., 2021; Chen et al., 2021; Zhu et al., 2022; Shu et al., 2021) breaks this paradigm by estimating the model quality with other metrics that are fast to compute. However, the search quality heavily relies on the effectiveness of the metrics.

Several prior works, such as Liu et al. (2018a), Li et al. (2020), Qian et al. (2022), have introduced progressive search mechanisms on layerwise search spaces. LayerNAS stands apart by explicitly articulating the underlying assumptions of the layerwise search space, rigorously deriving the method from these assumptions, and effectively constraining the polynomial search space complexity.

## 3 PROBLEM DEFINITION

Most NAS algorithms do not differentiate the various types of NAS problems. Rather, they employ a single encoding of the search space with a general solution for the search process. However, the unique characteristics of NAS problems can be leveraged to design a tailored approach. We categorize NAS problems into three major types:

- Topology search: the search space defines a graph with multiple nodes. The objective is to identify an optimal topology for connecting nodes with different operations. This task allows for the exploration of novel architectures.

- Size search or compression search: the search occurs on a predefined model architecture with multiple layers. Each layer can be selected from as a set of search options. Empirically, the best-performing model is normally the one with the most parameters per layer. Therefore, in practice, we aim to search for the optimal model under certain constraints. NATSBench size search (Dong et al., 2021) provides a public dataset for this type of task. MobilNetV3 (Howard et al., 2019), EfficientNet (Tan & Le, 2019), FBNet (Wu et al., 2019) also establish the search space in this manner. This problem can also be viewed as a compression problem He et al. (2018), as reducing the layer size serves as a means of compression by decreasing the model size, FLOPs and latency.

- Scale search: model architectures are uniformly configured by hyper-parameters, such as number of layers or size of fully-connected layers. This task views the model as a holistic entity and uniformly scales it up or down, rather than adjusting individual layers or components.

This taxonomy illustrates the significant variation among NAS problems. Rather than proposing a general solution to address all of them, we propose to tackle with search spaces in a layerwise manner. Specifically, we aim to find a model with $L$ layers. For each $layer_i$, we select from a set of search options $\mathbb{S}_i$. A model candidate $\mathcal{M}$ can be represented as a tuple with size $L$: $(s_1, s_2, ..., s_L)$. $s_i \in \mathbb{S}_i$ is a selected search option on $layer_i$. The objective is to find an optimal model architecture $\mathcal{M} = (s_1, s_2, ..., s_L)$ with the highest accuracy:

$$\underset{(s_1, s_2, ..., s_L)}{\mathrm{argmax}} \quad Accuracy(\mathcal{M}) \tag{1}$$

## 4 METHOD

We propose LayerNAS as an algorithm that leverages layerwise attributes. When searching models $\mathcal{M}_i$ on $layer_i$, we are searching for architectures in the form of $(s_{1..i-1}, x_i, o_{i+1..L})$. $s_{1..i-1}$ are the selected options for $layer_{1..i-1}$, and $o_{i+1..L}$ are the default, predefined options. $x_i$ is the search option selected for $layer_i$, which is the current layer in the search. In this formulation, only $layer_i$ can be changed, all preceding layers are fixed, and all succeeding layers are using the default option. In topology search, the default option is usually no-op; in size search, the default option can be the option with most computation.

LayerNAS operates on a search space that meets the following assumption, which has been implicitly utilized by past chain-structured NAS techniques (Liu et al., 2018a; Tan et al., 2019; Howard et al., 2019).

**Assumption 4.1.** The optimal model $\mathcal{M}_i$ on $layer_i$ can be constructed from a model $m \in \mathbb{M}_{i-1}$, where $\mathbb{M}_{i-1}$ is a set of model candidates on $layer_{i-1}$.

This assumption implies:

- Enforcing a sequential search process is possible when exploring $layer_i$ because improvements to the model cannot be achieved by modifying $layer_{i-1}$.

- The information for finding an optimal model architecture on $layer_i$ was collected when searching for model architectures on $layer_{i-1}$.

- Search spaces that are constructed in a layerwise manner, such as those in size search problems discussed in Section 3, can usually meet this assumption. Each search option can completely define how to construct a succeeding layer, and does not depend on the search options in previous layers.

- It's worth noting that not all search spaces can meet the assumption. Succeeding layers may be coupled with or affect preceding layers in some cases. In practice, we transform the search space in Section 5 to ensure that it meets the assumption.

---

**Algorithm 1** LayerNAS algorithm

---

    **Inputs:** $L$ (num layers), $R$ (num searches per layer), $T$ (num models to generate in next layer)
    $l = 1$
    $\mathbb{M}_1 = \{\forall \mathcal{M}_1\}$
    **repeat**
        **for** $i = 1$ to $R$ **do**
            $\mathcal{M}_l = \text{select}(\mathbb{M}_l)$
            **for** $j = 1$ to $T$ **do**
                $\mathcal{M}_{l+1} = \text{apply\_search\_option}(\mathcal{M}_l, \mathbb{S}_{l+1})$
                $h = \varphi(\mathcal{M}_{l+1})$
                $accuracy = \text{train\_and\_eval}(\mathcal{M}_{l+1})$
                **if** $accuracy > \text{Accuracy}(\mathbb{M}_{l+1,h})$ **then**
                    $\mathbb{M}_{l+1,h} = \mathcal{M}_{l+1}$
                **end if**
            **end for**
        **end for**
        $l = l + 1$
        **if** $l == L$ **then**
            $l = 1$
        **end if**
    **until** no available candidates

---

## 4.1 LayerNAS for Topology Search

The LayerNAS algorithm is described by the pseudo code in Algorithm 1. $\mathbb{M}_l$ is a set of model candidates on $\text{layer}_l$. $\mathbb{M}_{l,h}$ is the model on $\text{layer}_l$ mapped to $h \in \mathbb{H}$, a lower dimensional representation. $\mathbb{H}$ is usually a finite integer set, so that we can index and store models.

$\varphi : \mathbb{M} \to \mathbb{H}$ maps model architecture in $\mathbb{M}$ to a lower dimensional representation $\mathbb{H}$. $\varphi$ could be an encoded index of model architecture, or other identifiers that group similar model architectures. We discuss this further in 4.2. When there is a unique id for each model, LayerNAS will store all model candidates in $\mathbb{M}$.

In this algorithm, the total number of model candidates we need to search is $\sum_{i=1}^{L} |\mathbb{M}_i| \cdot |\mathbb{S}_i|$. It has a polynomial form, but $|\mathbb{M}_L| = O(|\mathbb{S}|^L)$ if we set $\varphi(\mathcal{M})$ as unique id of models. This does not limit the order of $|\mathbb{M}_L|$ to search. For topology search, we can design a sophisticated $\varphi$ to group similar model candidates. In the following discussion, we will demonstrate how to lower the order of $|\mathbb{M}_L|$ in multi-objective NAS.

## 4.2 LayerNAS for Multi-objective NAS

LayerNAS is aimed at designing an efficient algorithm for size search or compression search problems. As discussed in Section 3, such problems satisfy Assumption 4.1 by nature. Multi-objective NAS usually searches for an optimal model under some constraints, such as model size, inference latency, hardware-specific FLOPs or energy consumption. We use "cost" as a general term to refer to these constraints. These "cost" metrics are easy to calculate and can be determined when the model architecture is fixed. This is in contrast to calculating accuracy, which requires completing the model training. Because the model is constructed in a layer-wise manner, the cost of the model can be estimated by summing the costs of all layers.

Hence, we can express the multi-objective NAS problem as,

$$\underset{(s_1, s_2, \ldots, s_L)}{\text{argmax}} \quad Accuracy(\mathcal{M}_L)$$

$$\text{s.t.} \quad \sum_{i=1}^{L} Cost(s_i) \leq \text{target} \tag{2}$$

where $Cost(s_i)$ is the cost of applying option $s_i$ on $\text{layer}_i$.

We introduce an additional assumption by considering the cost in Assumption 4.1:

**Assumption 4.2.** The optimal model $\mathcal{M}_i$ with cost $= C$ when searching for $\text{layer}_i$ can be constructed from the optimal model $\mathcal{M}_{i-1}$ with cost $= C - Cost(s_i)$ from $\mathbb{M}_{i-1}$.

In this assumption, we only keep one optimal model out of a set of models with similar costs. Suppose we have two models with the same cost, but $\mathcal{M}_i$ has better quality than $\mathcal{M}'_i$. The assumption will be satisfied if any changes on following layers to $\mathcal{M}_i$ will generate a better model than making the same change to $\mathcal{M}'_i$.

By applying Assumption 4.2 to Equation (2), we can formulate the problem as combinatorial optimization:

$$
\begin{aligned}
\underset{x_i}{\text{argmax}} \quad & Accuracy(\mathcal{M}_i) \\
\text{s.t.} \quad & \sum_{j=1}^{i-1} Cost(s_{1..i-1}, x_i, o_{i+1,L}) \leq \text{target} \\
\text{where} \quad & \mathcal{M}_i = (s_{1..i-1}, x_i, o_{i+1..L}), \mathcal{M}_{i-1} = (s_{1..i-1}, o_{i..L}) \in \mathbb{M}_{i-1,h'}
\end{aligned}
\tag{3}
$$

This formulation decouples cost from reward, so there is no need to manually design an objective function to combine these metrics into a single value, and we can avoid tuning hyper-parameters of such an objective. Formulating the problem as combinatorial optimization allows solving it efficiently using dynamic programming. $\mathbb{M}_{l,h}$ can be considered as a memorial table to record best models on $\text{layer}_l$ at cost $h$. For $\text{layer}_l$, $\mathcal{M}_l$ generates the $\mathcal{M}_{l+1}$ by applying different options selected from $\mathbb{S}_{l+1}$ on $\text{layer}_{l+1}$. The search complexity is $O(H \cdot |\mathbb{S}| \cdot L)$.

We do not need to store all $\mathbb{M}_{l,h}$ candidates, but rather group them with the following transformation:

$$
\varphi(\mathcal{M}_i) = \left\lfloor \frac{Cost(\mathcal{M}_i) - \min Cost(\mathbb{M}_i)}{\max Cost(\mathbb{M}_i) - \min Cost(\mathbb{M}_i)} \times H \right\rfloor
\tag{4}
$$

where $H$ is the desired number of buckets to keep. Each bucket contains model candidates with costs in a specific range. In practice, we can set $H = 100$, meaning we store optimal model candidates within 1% of the cost range.

Equation (4) limits $|\mathbb{M}_i|$ to be a constant value since $H$ is a constant. $\min Cost(\mathbb{M}_i)$ and $\max Cost(\mathbb{M}_i)$ can be easily calculated when we know how to select the search option from $\mathbb{S}_{i+1}..\mathbb{S}_L$ in order to maximize or minimize the model cost. This can be achieved by defining the order within $\mathbb{S}_i$. Let $s_i = 1$ represent the option with the maximal cost on $\text{layer}_i$, and $s_i = |\mathbb{S}|$ represent the option with the minimal cost on $\text{layer}_i$. This approach for constructing the search space facilitates an efficient calculation of maximal and minimal costs.

The optimization applied above leads to achieving polynomial search complexity $O(H \cdot |\mathbb{S}| \cdot L)$. $O(|\mathbb{M}|) = H$ is upper bound of the number of model candidates in each layer, and becomes a constant after applying Equation (4). $|\mathbb{S}|$ is the number of search options on each layer.

LayerNAS for Multi-objective NAS does not change the implementation of Algorithm 1. With the same framework, we just need to set $\varphi$ to group $\mathbb{M}_i$ by their costs with Equation (4). In practice, Assumption 4.2 is not always true because accuracy may vary in each training trial. The algorithm may store a lucky model candidate that happens to get a better accuracy due to variation. We store multiple candidates for each $h$ to reduce the problem from training accuracy variation.

## 5 EXPERIMENTS

### 5.1 SEARCH ON IMAGENET

**Search Space:** we construct several search spaces based on MobileNetV2, MobileNetV2 (width multiplier=1.4), MobileNetV3-Small and MobileNetV3-Large. For each search space, we set similar backbone of the base model. For each layer, we consider kernel sizes from {3, 5, 7}, base filters and expanded filters from a set of integers, and a fixed strides. The objective is to find better models with similar MAdds of the base model.

To avoid coupling between preceding and succeeding layers, we first search the shared base filters in each block to create residual shortcuts, and search for kernel sizes and expanded filters subsequently. This ensures the search space satisfy Assumption 4.1.

We estimate and compare the number of unique model candidates defined by the search space and the maximal number of searches in Table 1. In the experiments, we set $H = 100$, and store 3 best models with same $h$-value. Note that the maximal number of searches does not mean actual searches conducted in the experiments, but rather an upper bound defined by the algorithm.

A comprehensive description of the search spaces and discovered model architectures in this experiment can be found in the Appendix for further reference.

Table 1: Comparison of model candidates in the search spaces

| Search Space | Target MAdds | # Unique Models | # Max Trials |
|---|---|---|---|
| MobileNetV3-Small | 60M | $5.0e + 20$ | $1.2e + 5$ |
| MobileNetV3-Large | 220M | $4.8e + 26$ | $1.5e + 5$ |
| MobileNetV2 | 300M | $5.3e + 30$ | $1.4e + 5$ |
| MobileNetV2 1.4x | 600M | $1.6e + 39$ | $2.0e + 6$ |

**Search, train and evaluation:**

During the search process, we train the model candidates for 5 epochs, and use the top-1 accuracy on ImageNet as a proxy metrics. Following the search process, we select several model architectures with best accuracy on 5 epochs, train and evaluate them on 4x4 TPU with 4096 batch size (128 images per core). We use RMSPropOptimizer with 0.9 momentum, train for 500 epochs. Initial learning rate is 2.64, with 12.5 warmup epochs, then decay with cosine schedule.

**Results**

We list the best models discovered by LayerNAS, and compare them with baseline models and results from recent NAS works in Table 6. For all targeted MAdds, the models discovered by LayerNAS achieve better performance: 69.0% top-1 accuracy on ImageNet for 61M MAdds, a 1.6% improvement over MobileNetV3-Small; 75.6% for 229M MAdds, a 0.4% improvement over MobileNetV3-Large; 77.1% accuracy for 322M MAdds, a 5.1% improvement over MobileNetV2; and finally, 78.6% accuracy for 627M MAdds, a 3.9% improvement over MobileNetV2 1.4x.

Note that for all of these models, we include squeeze-and-excitation blocks (Hu et al., 2018) and use Swish activation (Ramachandran et al., 2017), in order to to achieve the best performance. Some recent works on NAS algorithms, as well as the original MobileNetV2, do not use these techniques. For a fair comparison, we also list the model performance after removing squeeze-and-excitation and replacing Swish activation with ReLU. The results show that the relative improvement from LayerNAS is present even after removing these components.

### 5.2 NATS-BENCH

The following experiments compare LayerNAS with other NAS algorithms on NATS-Bench (Dong et al., 2021). We evaluate NAS algorithms from three perspectives:

- Candidate quality: the quality of the best candidate model found by the algorithm, i.e. the peak evaluation accuracy during search.
- Stability: the ability to find the best candidate, after running multiple searches and analyzing the average value and range of variation.
- Efficiency: The training time required to find the best candidate. The sooner the peak accuracy candidate is reached, the more efficient the algorithm.

**NATS-Bench topology search**

NATS-Bench topology search defines a search space on 6 ops that connect 4 tensors, each op has 5 options (conv1x1, conv3x3, maxpool3x3, no-op, skip). It contains 15625 candidates with

Table 2: Comparison of models on ImageNet

| Model | Top1 Acc. | Params | MAdds |
|---|---|---|---|
| MobileNetV3-Small (Howard et al., 2019)[†] | 67.4 | 2.5M | 56M |
| MNasSmall (Tan et al., 2019) | 64.9 | 1.9M | 65M |
| **LayerNAS** (Ours)[†] | **69.0** | 3.7M | 61M |
| MobileNetV3-Large (Howard et al., 2019)[†] | 75.2 | 5.4M | 219M |
| **LayerNAS** (Ours) [†] | **75.6** | 5.1M | 229M |
| MobileNetV2 (Sandler et al., 2018)[⋆] | 72.0 | 3.5M | 300M |
| ProxylessNas-mobile (Cai et al., 2018)[⋆] | 74.6 | 4.1M | 320M |
| MNasNet-A1 (Tan et al., 2019) | 75.2 | 3.9M | 315M |
| FairNAS-C (Chu et al., 2021)[⋆] | 74.7 | 5.6M | 325M |
| LayerNAS-no-SE(Ours)[⋆] | 75.5 | 3.5M | 319M |
| EfficientNet-B0 (Tan & Le, 2019) | 77.1 | 5.3M | 390M |
| SGNAS-B (Huang & Chu, 2021) | 76.8 | - | 326M |
| FairNAS-C (Chu et al., 2021)[†] | 76.7 | 5.6M | 325M |
| GreedyNAS-B (You et al., 2020)[†] | 76.8 | 5.2M | 324M |
| **LayerNAS** (Ours)[†] | **77.1** | 5.2M | 322M |
| MobileNetV2 1.4x (Sandler et al., 2018)[⋆] | 74.7 | 6.9M | 585M |
| ProgressiveNAS (Liu et al., 2018a)[⋆] | 74.2 | 5.1M | 588M |
| Shapley-NAS (Xiao et al., 2022)[⋆] | 76.1 | 5.4M | 582M |
| MAGIC-AT (Xu et al., 2022)[⋆] | 76.8 | 6M | 598M |
| LayerNAS-no-SE (Ours)[⋆] | 77.1 | 7.6M | 598M |
| **LayerNAS** (Ours) [†] | **78.6** | 9.7M | 627M |

[⋆] Without squeeze-and-excitation blocks.
[†] With squeeze-and-excitation blocks.

their number of parameters, FLOPs, accuracy on Cifar-10, Cifar-100 (Krizhevsky et al., 2009), ImageNet16-120 (Chrabaszcz et al., 2017). In Table 3, we compare with recent state-of-the-art methods. Although training-free NAS has advantage of lower search cost, LayerNAS can achieve much better results.

Table 3: Comparison on NATS-Bench topology search. Average test accuracy on 5 runs.

| | Cifar10 | Cifar100 | ImageNet16-120 | Cost (sec) |
|---|---|---|---|---|
| RS | 92.39±0.06 | 63.54±0.24 | 42.71±0.34 | 1e+5 |
| RE (Real et al., 2019) | 94.13±0.18 | 71.40±0.50 | 44.76±0.64 | 1e+5 |
| PPO (Schulman et al., 2017) | 94.02±0.13 | 71.68±0.65 | 44.95±0.52 | 1e+5 |
| KNAS (Xu et al., 2021) | 93.05 | 68.91 | 34.11 | 4200 |
| TE-NAS (Chen et al., 2021) | 93.90±0.47 | 71.24±0.56 | 42.38±0.46 | 1558 |
| EigenNas (Zhu et al., 2022) | 93.46±0.02 | 71.42±0.63 | 45.54±0.04 | - |
| NASI (Shu et al., 2021) | 93.55±0.10 | 71.20±0.14 | 44.84±1.41 | 120 |
| FairNAS (Chu et al., 2021) | 93.23±0.18 | 71.00±1.46 | 42.19±0.31 | 1e+5 |
| SGNAS (Huang & Chu, 2021) | 93.53±0.12 | 70.31±1.09 | 44.98±2.10 | 9e+4 |
| LayerNAS | **94.34±0.12** | **73.01±0.63** | **46.58±0.59** | 1e+5 |
| Optimal test accuracy | 94.37 | 73.51 | 47.31 | |

**NATS-Bench size search**

NATS-Bench size search defines a search space on a 5-layer CNN model, each layer has 8 options on different number of channels, from 8 to 64. The search space contains 32768 model candidates. The one with the highest accuracy has 64 channels for all layers, we can refer this candidate as "the largest model". Instead of searching for the best model, we set the goal to search for the optimal model with 50% FLOPs of the largest model.

Under this constraints for size search, we implement popular NAS algorithms for comparison, which are also used in the original benchmark papers (Ying et al., 2019; Dong et al., 2021): random search, proximal policy optimization (PPO) (Schulman et al., 2017) and regularized evolution (RE) (Real

et al., 2019). We conduct 5 runs for each algorithm, and record the best accuracy at different training costs.

LayerNAS treats this as a compression problem. The base model, which is the largest model, has 64 channels on all layers. By applying search options with fewer channels, the model becomes smaller, faster and less accurate. The search process is to find the optimal model with expected FLOPs. By filtering out candidates that do not produce architectures falling within the expected FLOPs range, we can significantly reduce the number of candidates that need to be searched.

Table 4: Comparison on NATS-Bench size search. Average test accuracy on 5 runs.

|  | Cifar10 | Cifar100 | ImageNet16-120 |
|---|---|---|---|
| Training time (sec) | 2e+5 | 4e+5 | 6e+5 |
| Target mFLOPs | 140 | 140 | 35 |
| RS | 0.9265 | 0.6935 | 0.4381 |
| RE (Real et al., 2019) | 0.9282 | 0.6962 | 0.4476 |
| PPO (Schulman et al., 2017) | 0.9283 | 0.6957 | 0.4438 |
| LayerNAS | **0.9320** | **0.7064** | **0.4537** |
| Optimal validation | 0.9264 | 0.6922 | 0.4500 |
| Optimal test | 0.9334 | 0.7086 | 0.4553 |

## 6 CONCLUSION AND FUTURE WORK

In this research, we propose LayerNAS that formulates Multi-objective Neural Architecture Search to Combinatorial Optimization. By decoupling multi-objectives into cost and accuracy, and leverages layerwise attributes, we are able to reduce the search complexity from $O(|\mathbb{S}|^L)$ to $O(H \cdot |\mathbb{S}| \cdot L)$.

Our experiment results demonstrate the effectiveness of LayerNAS in discovering models that achieve superior performance compared to both baseline models and models discovered by other NAS algorithms under various constraints of MAdds. Specifically, models discovered through LayerNAS achieve top-1 accuracy on ImageNet of 69% for 61M MAdds, 75.6% for 229M MAdds, 77.1% for 322M MAdds, 78.6% for 627M MAdds. Furthermore, our analysis reveals that LayerNAS outperforms other NAS algorithms on NATS-Bench in all aspects including best model quality, stability and efficiency.

While the current implementation of LayerNAS has shown promising results, several current limitations that can be addressed by future work:

- LayerNAS is not designed to solve scale search problems mentioned in Section 3, because many hyper-parameters of model architecture are interdependent in scale search problem, which contradicts the statement in Assumption 4.1.
- One-shot NAS algorithms have been shown to be more efficient. We aim to investigate the potential of applying LayerNAS to One-shot NAS algorithms.

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

## A    NOTATION

$\mathbb{S}_i$: Search options for $layer_i$.

$|\mathbb{S}_i|$: Num of search options on $layer_i$.

$s_i$: selected search option on $layer_i$ from the set $\mathbb{S}_i$

$o_i$: default search option applied on $layer_i$, $o_{1..L}$ is the architecture of the baseline model.

$(s_1, s_2, .., s_L)$: A model architecture that applies $s_1$ on $layer_1$, $s_2$ on $layer_2$, ... , $s_L$ on $layer_L$

$(s_{1..i-1}, x_i, o_{i+1..L})$: A model architecture that applies $s_1$ on $layer_1$, $s_2$ on $layer_2$, ..., default search option $o_{i+1}$ on $layer_{i+1}$, ... $o_L$ on $layer_L$, and search $x_i$ on $layer_i$.

$\mathcal{M}_i$: A model candidate that is searched on $layer_i$, it's in the form of $(s_{1..i-1}, x_i, o_{i+1..L})$.

$\mathbb{M}_i$: All model candidates searching on $layer_i$.

$\mathbb{M}_{i,h}$: Model candidates searching on $layer_i$, and are mapped to $h \in \mathbb{H}$.

$\varphi : \mathbb{M} \to \mathbb{H}$: transforms a model architecture $\mathcal{M} \in \mathbb{M}$ to a finite integer set $\mathbb{H}$

## B    NASBENCH-101 SEARCH DETAILS

NASBench-101 defines a search space on 5 ops, each op has 3 options (conv1x1, conv3x3, maxpool 3x3), and 21 potential edges to connect these ops and input, output ops. It contains 509M candidates with their number of parameters, accuracy on Cifar-10, and other information.

We construct the LayerNAS search space by adding a new edge for each layer. Search options in each layer are used to determine either to include a new op or connect two existing ops. By doing so, all constructed candidates can be legit, because all candidates are connected graphs. And this approach of search space construction can satisfy the assumption of LayerNAS: the best model candidate in $layer_i$ can be constructed from candidates in $layer_{i-1}$ by adding an new edge.

In the experiments, Regularized Evolution (RE) sets population_size=50, tournament_size=10; Proximal Policy Optimization (PPO) sets train_batch_size=16, update_batch_size=8, num_updates_per_feedback=10. Both RE and PPO are using MNAS as objective function: $Accuracy \times (Cost/Target)^{-0.07}$

In Figure 3, we observe that in earlier searching iterations LayerNAS performs slightly worse than other algorithms. This is because LayerNAS initially searches model candidates with fewer ops and edges, which intuitively perform poorly. However, after collecting enough information from early layers, LayerNAS consistently performs better. This is because LayerNAS does not rely on randomness, rather, it adds ops and edges from successful candidates in each layers, leading to continuous improvement.

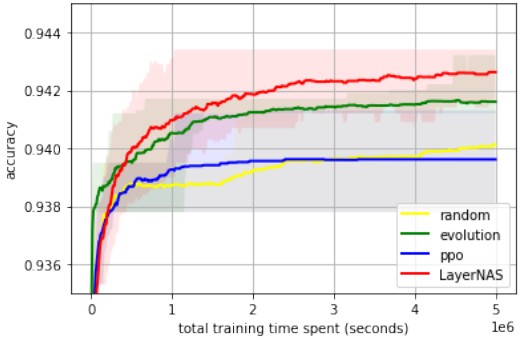

Figure 3: NASBench-101 test accuracy on Cifar-10, average on 100 runs

Table 5: Comparison on NASBench-101

| Algorithm | Validation accuracy | Test accuracy |
|-----------|--------------------|--------------| 
| RS | 0.9480 | 0.9401 |
| RE | 0.9497 | 0.9416 |
| PPO | 0.9476 | 0.9396 |
| LayerNAS | **0.9505** | **0.9426** |
| Optimal | 0.9432 | 0.9445 |

## C  NATS-BENCH SEARCH DETAILS

In the experiments, Regularized Evolution (RE) sets population_size=50, tournament_size=10; Proximal Policy Optimization (PPO) sets train_batch_size=16, update_batch_size=8, num_updates_per_feedback=10. Both RE and PPO are using MNAS as objective function: $Accuracy \times (Cost/Target)^{-0.07}$

### C.1  NATS-BENCH TOPOLOGY SEARCH

NATS-Bench topology search defines a search space on 6 ops that connect 4 tensors, each op has 5 options (conv1x1, conv3x3, maxpool3x3, no-op, skip).

In our experiments, we construct the LayerNAS search space by adding a new tensor for each layer. Search options in each layer are encoded with all op types that connect this tensor to previous tensors. So it has only 3 layers, each layer has 5, 25, 125 options.

Validation and test accuracy are shown in Figure 4 and Figure 5.

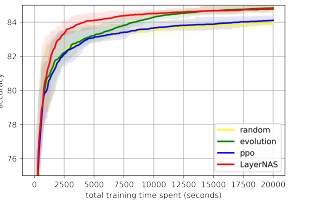 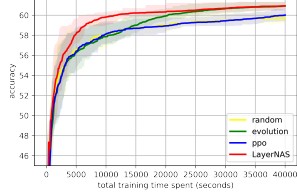 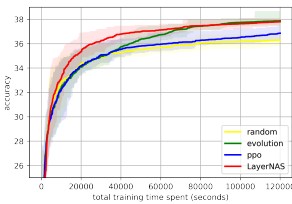

Figure 4: NATS-Bench topology search valid accuracy on (a) Cifar10 (b) Cifar100 (c) Imagenet16-120

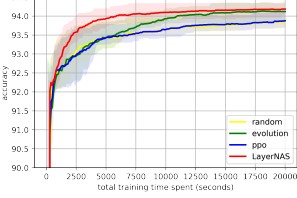 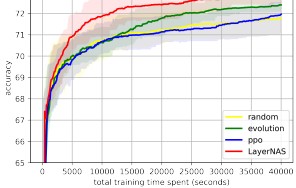 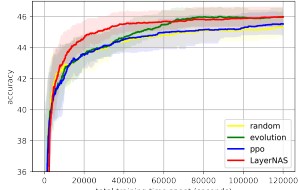

Figure 5: NATS-Bench topology search test accuracy on (a) Cifar10 (b) Cifar100 (c) Imagenet16-120

### C.2  NATS-BENCH SIZE SEARCH

NATS-Bench size search provides a dataset with information on model architectures with 5 layers. Each layer is a convolutional layer with different num of channels selected from {8, 16, 24, 32, 40,

48, 56, 64}. The model with 64 channels for all layers has the most model parameters, the largest latency and the best accuracy. The objective is to find the optimal model with 50% FLOPs.

LayerNAS constructs the search space by using the largest model as base model, and applies search options that reduce channels per layer. Althoughh LayerNAS steadily improves valid accuracy over time, test accuracy drops. This is due to in-correlation between test accuracy and valid accuracy.

Validation and test accuracy are shown in Figure 6 and Figure 7. We can observe that LayerNAS can outperform other algorithms on both validation and test accuracy. We can also attribute test accuracy drop in LayerNAS to the lack of correlation with validation accuracy.

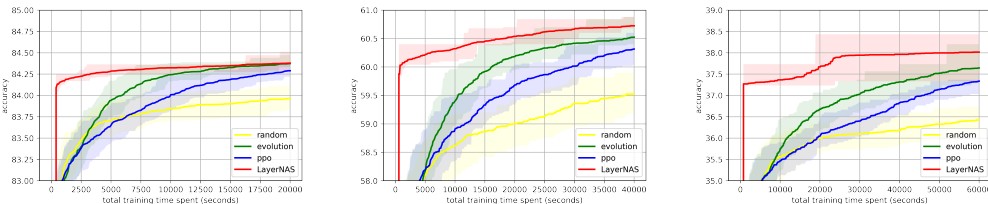

Figure 6: NATS-Bench size search valid accuracy on (a) Cifar10 (b) Cifar100 (c) Imagenet16-120

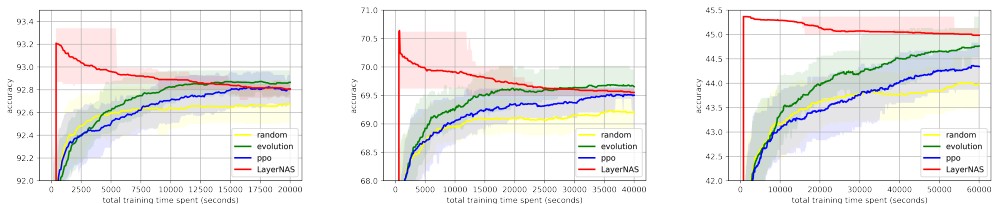

Figure 7: NATS-Bench size search test accuracy on (a) Cifar10 (b) Cifar100 (c) Imagenet16-120

## D  DYNAMIC PROGRAMMING IMPLEMENTATION OF LAYERNAS FOR MULTI-OBJECTIVE NAS

Algorithm 2 demonstrates how to implement LayerNAS with Dynamic Programming, which has clear explanation why search complexity is $O(H \cdot |\mathbb{S}| \cdot L)$ .

The implementation is not used in practice because it spends most of time searching in layer$_{1..L-1}$, we cannot get a model in expected cost range until last layer is searched.

---

**Algorithm 2** Dynamic Programming for Combinatorial Optimization

---

    **for** $l = 1$ to $L - 1$ **do**
      **for** $\mathcal{M}_l \in \mathbb{M}_l$ **do**
        **for** $s \in \mathbb{S}_{l+1}$ **do**
          $\mathcal{M}_{l+1}$ = apply_search_option($\mathcal{M}_l$, $s$)
          $h$ = cost($\mathcal{M}_{l+1}$)
          $accuracy$ = train_and_eval($\mathcal{M}_{l+1}$)
          **if** $accuracy >$ Accuracy($\mathbb{M}_{l+1,h}$) **then**
            $\mathbb{M}_{l+1,h} = \mathcal{M}_{l+1}$
          **end if**
        **end for**
      **end for**
    **end for**

---

# E    DISCUSSION ON SEARCH SPACE ASSUMPTIONS

Assumption 4.1 sets some characteristics of search spaces that can be leveraged to improve the search efficiency. Instead of expecting all search spaces can satisfy this assumption, in experiments, we construct search spaces based on MobileNet to intentionally make them satisfy Assumption 4.1. While we cannot guarantee that all search spaces can be transformed to satisfy Assumption 4.1, most search spaces used in existing models or studies either implicitly use this assumption or can be transformed to satisfy it. We also demonstrate the effectiveness of this assumption from the experiments on MobileNet.

## E.1    SEARCH SPACE IS COMPLETE

Assume we are searching for the optimal model $s_1..s_n$, and we store all possible model candidates on each layer. During the search process on $\text{layer}_n$, we generate model architectures by modifying $o_n$ to other options in $\mathbb{S}$. Since we store all model architectures for $\text{layer}_{n-1}$, the search process can create all $|\mathbb{S}|^n$ candidates on $\text{layer}_n$ by adding each $s_n \in \mathbb{S}$ to the models in $\mathbb{M}_{n-1}$. Therefore, $\mathbb{M}_n$ contains all possibilities in the search space. This process can then be applied backward to the first layer.

## E.2    SEQUENTIAL SEARCH ORDER

Assume, after LayerNAS sequential search, we get optimal model defined as $a_1..a_i..a_n$. For sake of contradiction, there exists a model $a_1..b_i..a_n$, with superior performance, by applying a change in previous layers. Since the search space is complete, model $a_1..b_i o_{i+1}..o_n$ must exist, and has been processed in $\mathbb{M}_i$. In the sequential search, model $a_1..b_i a_{i+1}..o_n$ can be created by using $a_{i+1}$ on $\text{layer}_{i+1}$. Repeating this process for all subsequent layers will eventually lead to $a_1..b_i..a_n$, contradicting our assumption that optimal model from sequential search is $a_1..a_i..a_n$. Therefore, we can search sequentially.

## E.3    LIMIT OF THE ASSUMPTION

MobileNet architecture does not satisfy Assumption 4.1 by default. Residual requires $\text{layer}_i$ and $\text{layer}_j$ have the same num of filters. Suppose $\mathbb{S}_i = \{32, 64, 96\}$, $\mathbb{S}_j = \{64, 96, 128\}$, the residual shortcut cannot be created if $s_i = 32, s_j = 96$. This is the case when preceding layers are coupled with succeeding layers. To overcome this issue, we introduce a virtual layer, with options $\{64, 96\}$. We first search this shared filter to create residual shortcuts, and then search specs for each layer. This transformation ensures that the new search space satisfy Assumption 4.1. In the case of MobileNet search space, we first search for the common filters for the block and then for the expanded filters for each layer. This approach allows us to perform LayerNAS on a search space that satisfies Assumption 4.1.

# F    DISCUSSION ON NUM OF REPLICAS TO STORE

From experiments on MobileNet, we observed that multiple runs on the same model architecture can yield standard deviations of accuracy ranging from 0.08% to 0.15%. Often times, the difference can be as high as 0.3%. To address this, we propose storing multiple candidates for the same cost to increase the likelihood of keeping the better model architecture for every layer search.

Suppose we have two models with the same cost, $x$ and $y$, where $x$ is inferior and $y$ is superior, and the training accuracy follows a Gaussian distribution $N(\mu, \sigma^2)$. The probability of $x$ obtaining a higher accuracy than $y$ is $P(x - y > 0)$, where $x - y \sim N(\mu_x - \mu_y, \sigma_x{}^2 + \sigma_y{}^2)$. In emprical examples, $\mu_x - \mu_y = -0.002$ and $\sigma_x = 0.001$, then $x$ has the probability of 9.2% of obtaining a higher accuracy. When we have $L = 20$ layers, the probability of keeping the better model architecture for every layer search is $(1 - p)^{20} = 18\%$.

By storing $k$ candidates with the same cost, we can increase the probability of keeping the better model architecture. When $k = 3$, the probability of storing all inferior models is $p^k = 0.08\%$. The probability of keeping the better model architecture for all $L = 20$ layer searches is 98.4%, which is practically good enough.

Theoretically, if we store infinite candidates per layer, we are performing a complete grid search, which guarantees a optimal model architecture.

## G    TRANSFERABILITY

LayerNAS's explored model architectures exhibit improved performance across various tasks as well.

Table 6: Comparison of models on ImageNet

| Model | ImageNet top-1 acc | CoCo mAP | Params | MAdds |
|---|---|---|---|---|
| MobileNetV2 | 72.0 | 22.1 | 3.5M | 300M |
| LayerNAS w/o SE | 77.1 | 23.85 | 7.6M | 598M |
| LayerNAS | 78.6 | 24.84 | 9.7M | 527M |
| MobileNetV3-Small | 67.4 | 16 | 2.5M | 56M |
| LayerNAS | 69.0 | 17.94 | 3.7M | 61M |
| MobileNetV3-Large | 75.2 | 22.0 | 5.4M | 219M |
| LayerNAS | 75.6 | 23.75 | 5.1M | 229M |

## H    MOBILENETV2 AND MOBILENETV3 SEARCH DETAILS

We aim to search models under different MAdds constrants: 60M (similar to MobileNetV3-Small), 220M (similar to MobileNetV3-Large), 300M (similar to MobileNetV2), 600M (similar to MobileNetV2 1.4x).

For each block, we will search the number of output filters of the block first. All layers in the block have the same number of output filters to create residual block correctly. Following the search for the block output filters, we search expanded filter and kernel size of each layers in this block. Strides are fixed for all layers. We use $|\mathbb{S}|$ to denote the number of search options of this layer, which facilitates the computation on the number of unique model architectures, and max number of required search trials in LayerNAS.

### H.1    60M MADDS MODEL

The search spaces has $L = 16$ encoded length. Number of unique model architecture is $\prod |\mathbb{S}| = 5.0e + 20$. We store up to 300 model candidates per layer, so max number of trials is $300 \times \sum |\mathbb{S}| = 1.2e + 5$.

### H.2    220M MADDS MODEL

The search spaces has $L = 21$ encoded length, the number of unique model architecture is $\prod |\mathbb{S}| = 4.8e + 26$ For LayerNAS, we store up to 300 model candidates per layer, so max number of trials is $300 \times \sum |\mathbb{S}| = 1.5e + 5$

### H.3    300M MADDS MODEL

The search spaces has $L = 26$ encoded length, the number of unique model architectures is $\prod |\mathbb{S}| = 5.3e + 30$. We store up to 300 model candidates per layer, so max number of trials is $300 \times \sum |\mathbb{S}| = 1.4e + 5$.

### H.4    600M MADDS MODEL

The search spaces has $L = 31$ encoded length, the number of unique model architecture is $\prod |\mathbb{S}| = 1.6e + 39$ For LayerNAS, we store up to 300 model candidates per layer, so max number of trials is $300 \times \sum |\mathbb{S}| = 2.0e + 6$

Table 7: 60M MAdds Search Space

| Operator | # Output filter | # Expanded Filter | strides | $|\mathbb{S}|$ |
|---|---|---|---|---|
| conv2d{3x3} | 16 | | 2 | |
| bneck {3x3} | {24, 20, 18, 16, 14, 12} | | 2 | 6 |
| Block filter | {36, 32, 28, 24, 20, 18, 16} | | | 7 |
| bneck {3x3, 5x5} | | {144, 136, 128, 120, 112, 104, 96, 88, 80, 72, 68, 64, 60, 56} | 2 | 28 |
| bneck {3x3, 5x5} | | {144, 136, 128, 120, 112, 104, 96, 88, 80, 72 68, 64, 60, 56} | 1 | 28 |
| Block filter | {60, 56, 52, 48, 44, 40, 36, 32, 28} | | | 9 |
| bneck {3x3, 5x5, 7x7} | | {192, 176, 160, 144, 128, 112, 104, 96, 88, 80, 72, 64} | 2 | 36 |
| bneck {3x3, 5x5, 7x7} | | {480, 440, 400, 360, 320, 300, 280, 260, 240, 220, 200, 180, 160} | 1 | 39 |
| bneck {3x3, 5x5, 7x7} | | {480, 440, 400, 360, 320, 300, 280, 260, 240, 220, 200, 180, 160} | 1 | 39 |
| Block filter | {96, 88, 80, 72, 64, 60, 56, 52, 48, 44, 40, 36, 32} | | | 13 |
| bneck {3x3, 5x5, 7x7} | | {240, 200, 180, 160, 140, 120, 100, 90, 80} | 1 | 27 |
| bneck {3x3, 5x5, 7x7} | | {288, 256, 224, 208, 192, 176, 160, 152, 144, 136, 128, 120} | 1 | 36 |
| Block filter | {192, 176, 160, 144, 128, 120, 112, 104, 96, 88, 80, 72, 64} | | | 13 |
| bneck {3x3, 5x5, 7x7} | | {576, 544, 512, 480, 448, 416, 384, 352, 320, 288, 256, 224} | 2 | 36 |
| bneck {3x3, 5x5, 7x7} | | {1152, 1088, 1024, 960, 896, 832, 768, 704, 640, 576, 516, 448} | 1 | 36 |
| bneck {3x3, 5x5, 7x7} | | {1152, 1088, 1024, 960, 896, 832, 768, 704, 640, 576, 516, 448} | 1 | 36 |
| conv2d 1x1 | {864, 576}, | | | 2 |
| pool, 7x7 | | | | |
| conv2d 1x1 | {1536, 1024} | | | 2 |
| conv2d 1x1 | {1001} | | | |

Table 8: LayerNAS Model under 60M MAdds

| Input | Operator | # Output filter | # Expanded Filter | strides |
|---|---|---|---|---|
| $224 \times 224 \times 3$ | conv2d 3x3 | 16 | | 2 |
| $112 \times 112 \times 16$ | bneck 3x3 | 16 | | 2 |
| $56 \times 56 \times 16$ | bneck 3x3 | 28 | 144 | 2 |
| $28 \times 28 \times 28$ | bneck 3x3 | 28 | 128 | 1 |
| $28 \times 28 \times 28$ | bneck 5x5 | 44 | 96 | 2 |
| $14 \times 14 \times 44$ | bneck 3x3 | 44 | 220 | 1 |
| $14 \times 14 \times 44$ | bneck 3x3 | 44 | 200 | 1 |
| $14 \times 14 \times 44$ | bneck 7x7 | 40 | 160 | 1 |
| $14 \times 14 \times 40$ | bneck 3x3 | 40 | 152 | 1 |
| $14 \times 14 \times 96$ | bneck 5x5 | 96 | 224 | 2 |
| $7 \times 7 \times 96$ | bneck 3x3 | 96 | 448 | 1 |
| $7 \times 7 \times 96$ | bneck 3x3 | 96 | 512 | 1 |
| $7 \times 7 \times 96$ | conv2d 1x1 | 864 | | 1 |
| $7 \times 7 \times 864$ | pool, 7x7 | | | 1 |
| $7 \times 7 \times 864$ | conv2d 1x1 | 1536 | | 1 |
| $7 \times 7 \times 1536$ | conv2d 1x1 | 1001 | | 1 |

Table 9: 220M MAdds Search Space

| Operator | # Output filter | # Expanded Filter | strides | $|\mathbb{S}|$ |
|---|---|---|---|---|
| Conv2d{3x3} | 16 | | 2 | |
| bneck {3x3} | {24, 20, 18, 16, 14, 12} | | 1 | 6 |
| Block filter | {36, 32, 28, 24, 20, 16} | | | |
| bneck {3x3, 5x5} | | {96, 88, 80, 72, 68, 64, 60, 56, 48} | 2 | 18 |
| bneck {3x3, 5x5, 7x7} | | {124, 116, 108, 100, 92, 84, 72, 68, 64, 56, 48} | 1 | 33 |
| Block filter | {64, 56, 52, 48, 44, 40, 36, 32, 24} | | | 9 |
| bneck {3x3, 5x5, 7x7} | | {128, 120, 112, 104, 96, 88, 80, 76, 72, 64, 56} | 2 | 33 |
| bneck {3x3, 5x5, 7x7} | | {240, 200, 180, 160, 140, 120, 110, 100, 80} | 1 | 27 |
| bneck {3x3, 5x5, 7x7} | | {240, 200, 180, 160, 140, 120, 110, 100, 80} | 1 | 27 |
| Block filter | {160, 140, 130, 120, 110, 100, 80, 70, 60} | | | 9 |
| bneck {3x3, 5x5, 7x7} | | {360, 320, 300, 280, 260, 240, 220, 200, 180, 160} | 2 | 30 |
| bneck {3x3, 5x5, 7x7} | | {400, 360, 340, 320, 300, 280, 260, 240, 220, 200, 180, 160, 120} | 1 | 36 |
| bneck {3x3, 5x5, 7x7} | | {368, 336, 304, 288, 272, 256, 240, 224, 208, 184, 168, 152} | 1 | 36 |
| bneck {3x3, 5x5, 7x7} | | {368, 336, 304, 288, 272, 256, 240, 224, 208, 184, 168, 152} | 1 | 36 |
| Block filter | {224, 208, 192, 176, 160, 144, 128, 112, 96, 80} | | | 10 |
| bneck {3x3, 5x5, 7x7} | | {960, 880, 800, 720, 640, 560, 520, 480, 440, 400, 360} | 1 | 33 |
| bneck {3x3, 5x5, 7x7} | | {1344, 1200, 1056, 960, 888, 816, 768, 720, 624, 576, 480} | 1 | 33 |
| Block filter | {320, 280, 240, 220, 200, 180, 160, 120, 100 } | | | 9 |
| bneck {3x3, 5x5, 7x7} | | {1344, 1200, 1056, 960, 888, 816, 768, 720, 624, 576, 480} | 2 | 33 |
| bneck {3x3, 5x5, 7x7} | | {1920, 1760, 1600, 1440, 1280, 1120, 960, 880, 800, 720, 640} | 1 | 33 |
| bneck {3x3, 5x5, 7x7} | | {1920, 1760, 1600, 1440, 1280, 1120, 960, 880, 800, 720, 640} | 1 | 33 |
| bneck {3x3, 5x5, 7x7} | {480, 440, 400, 360, 320, 300, 280} | {1728, 1664, 1600, 1536, 1440, 1280, 1216} | 1 | 7 |
| conv2d 1x1 | {960} | | | |
| pool, 7x7 | | | | |
| conv2d 1x1 | {1440, 1280} | | | 2 |
| conv2d 1x1 | {1001} | | | |

Table 10: LayerNAS Model under 220M MAdds

| Input | Operator | # Output filter | # Expanded Filter | strides |
|---|---|---|---|---|
| $224 \times 224 \times 3$ | conv2d 3x3 | 16 | | 2 |
| $112 \times 112 \times 16$ | bneck 3x3 | 18 | | 1 |
| $112 \times 112 \times 16$ | bneck 3x3 | 24 | 64 | 2 |
| $56 \times 56 \times 28$ | bneck 3x3 | 24 | 48 | 1 |
| $56 \times 56 \times 28$ | bneck 5x5 | 56 | 80 | 2 |
| $28 \times 28 \times 44$ | bneck 5x5 | 56 | 200 | 1 |
| $28 \times 28 \times 44$ | bneck 5x5 | 56 | 100 | 1 |
| $28 \times 28 \times 44$ | bneck 5x5 | 80 | 400 | 2 |
| $14 \times 14 \times 40$ | bneck 3x3 | 80 | 200 | 1 |
| $14 \times 14 \times 96$ | bneck 3x3 | 80 | 272 | 1 |
| $7 \times 7 \times 96$ | bneck 3x3 | 80 | 168 | 1 |
| $14 \times 14 \times 44$ | bneck 5x5 | 112 | 440 | 1 |
| $14 \times 14 \times 40$ | bneck 5x5 | 112 | 576 | 1 |
| $14 \times 14 \times 96$ | bneck 7x7 | 160 | 624 | 2 |
| $7 \times 7 \times 96$ | bneck 5x5 | 160 | 640 | 1 |
| $7 \times 7 \times 96$ | bneck 3x3 | 160 | 640 | 1 |
| $7 \times 7 \times 96$ | conv2d 1x1 | 960 | | 1 |
| $7 \times 7 \times 864$ | pool, 7x7 | | | 1 |
| $7 \times 7 \times 864$ | conv2d 1x1 | 1280 | | 1 |
| $7 \times 7 \times 1536$ | conv2d 1x1 | 1001 | | 1 |

## I EXAMPLE

Consider a model with 3 layers, each with 4 options: A, B, C, D, corresponding to computational costs of 1M, 2M, 3M, and 4M MAdds, respectively. A sample model architecture can be represented as BCA, indicating that the 1st layer uses B, the 2nd layer uses C, and the 3rd layer uses A. The total cost of this model is 2+3+1=6M MAdds. The goal is to search for the optimal model architecture within the cost range of 8-9M MAdds.

LayerNAS settings:

- For each layer, candidates are grouped into 4 buckets, each bucket stores up to 2 candidates.

- In each iteration, 2 candidates are randomly selected to generate 2 valid children.

Cost range:

- 1st layer cost range: 9-12M, buckets: [9M], [10M], [11M], [12M]

- 2nd layer cost range: 6-12M, buckets: [6-7M], [8M], [9M, 10M], [11M, 12M]

- 3rd layer: only stores [8M, 9M]

Marks:

- Candidates that fall outside the designated cost range are marked as "drop"

- Once all child architectures have been generated, the model is marked with "[x]"

1st layer: train and eval: ADD (9M, 0.3), CDD (11M, 0.45)

2nd layer: Choose ADD and CDD
   ADD generates ABD (7M drop), ACD (8M, 0.27), AAD(6M drop), all children searched
   CDD generates CAD(8M, 0.4), CBD(9M, 0.42)
   [6-7M]: []
   [8M]: [ACD(8M, 0.27), CAD(8M, 0.4)]
   [9-10M]: [ADD(9M, 0.3)]

Table 11: 300M MAdds Search Space

| Operator | # Output filter | # Expanded Filter | strides | $|\mathbb{S}|$ |
|---|---|---|---|---|
| Conv2d{3x3} | 32 | | 2 | |
| bneck {3x3} | {24, 20, 16, 14} | | 1 | 4 |
| Block filter | {48, 44, 40, 36, 32, 28, 24} | | | 7 |
| bneck {3x3, 5x5} | | {72, 64, 56, 52, 48, 44, 40} | 2 | 14 |
| bneck {3x3, 5x5} | | {144, 128, 120, 112, 104, 96, 92, 88, 80, 76} | 1 | 20 |
| bneck {3x3, 5x5} | | {144, 128, 120, 112, 104, 96, 92, 88, 80, 76} | 1 | 20 |
| Block filter | {60, 56, 52, 48, 44, 40, 36, 32} | | | 8 |
| bneck {3x3, 5x5} | | {144, 128, 120, 112, 104, 96, 92, 88, 80, 76} | 2 | 20 |
| bneck {3x3, 5x5, 7x7} | | {180, 160, 140, 130, 120, 110, 100, 80} | 1 | 24 |
| bneck {3x3, 5x5, 7x7} | | {180, 160, 140, 130, 120, 110, 100, 80} | 1 | 24 |
| bneck {3x3, 5x5, 7x7} | | {180, 160, 140, 130, 120, 110, 100, 80} | 1 | 24 |
| Block filter | {120, 110, 100, 90, 80, 70, 60} | | | 7 |
| bneck {3x3, 5x5, 7x7} | | {360, 320, 280, 260, 240, 220, 200, 180} | 2 | 24 |
| bneck {3x3, 5x5, 7x7} | | {360, 320, 280, 260, 240, 220, 200, 180} | 1 | 24 |
| bneck {3x3, 5x5, 7x7} | | {360, 320, 280, 260, 240, 220, 200, 180} | 1 | 24 |
| Block filter | {144, 128, 120, 104, 96, 88, 80, 72} | | | 8 |
| bneck {3x3, 5x5, 7x7} | | {360, 320, 280, 260, 240, 220, 200, 180} | 1 | 24 |
| bneck {3x3, 5x5, 7x7} | | {432, 400, 368, 336, 304, 288, 272, 256, 240} | 1 | 27 |
| bneck {3x3, 5x5, 7x7} | | {432, 400, 368, 336, 304, 288, 272, 256, 240} | 1 | 27 |
| bneck {3x3, 5x5, 7x7} | | {432, 400, 368, 336, 304, 288, 272, 256, 240} | 1 | 27 |
| Block filter | {288, 256, 224, 192, 160, 144} | | | 6 |
| bneck {3x3, 5x5, 7x7} | | {864, 800, 736, 672, 608, 576, 512, 448} | 2 | 24 |
| bneck {3x3, 5x5, 7x7} | | {864, 800, 736, 672, 608, 576, 512, 448} | 1 | 24 |
| bneck {3x3, 5x5, 7x7} | | {864, 800, 736, 672, 608, 576, 512, 448} | 1 | 24 |
| bneck {3x3, 5x5, 7x7} | | {864, 800, 736, 672, 608, 576, 512, 448} | 1 | 24 |
| bneck {3x3, 5x5, 7x7} | {480, 440, 400, 360, 320, 300, 280} | {1728, 1664, 1600, 1536, 1440, 1280, 1216} | 1 | 7 |
| pool, 7x7 | | | | |
| conv2d 1x1 | {1920, 1600, 1280} | | | 3 |
| conv2d 1x1 | {1001} | | | |

Table 12: LayerNAS Model under 300M MAdds

| Input | Operator | # Output filter | # Expanded Filter | strides |
|---|---|---|---|---|
| $224 \times 224 \times 3$ | conv2d 3x3 | 32 | | 2 |
| $112 \times 112 \times 32$ | bneck 3x3 | 24 | | 1 |
| $112 \times 112 \times 24$ | bneck 3x3 | 28 | 40 | 2 |
| $56 \times 56 \times 28$ | bneck 3x3 | 28 | 144 | 1 |
| $56 \times 56 \times 28$ | bneck 3x3 | 28 | 88 | 1 |
| $56 \times 56 \times 28$ | bneck 3x3 | 40 | 104 | 2 |
| $28 \times 28 \times 40$ | bneck 5x5 | 40 | 110 | 1 |
| $28 \times 28 \times 40$ | bneck 3x3 | 40 | 180 | 1 |
| $28 \times 28 \times 40$ | bneck 5x5 | 40 | 130 | 1 |
| $28 \times 28 \times 40$ | bneck 7x7 | 90 | 260 | 2 |
| $14 \times 14 \times 90$ | bneck 3x3 | 90 | 220 | 1 |
| $14 \times 14 \times 90$ | bneck 3x3 | 90 | 200 | 1 |
| $14 \times 14 \times 90$ | bneck 7x7 | 120 | 320 | 1 |
| $14 \times 14 \times 120$ | bneck 5x5 | 120 | 288 | 1 |
| $14 \times 14 \times 120$ | bneck 7x7 | 120 | 256 | 1 |
| $14 \times 14 \times 120$ | bneck 3x3 | 120 | 368 | 1 |
| $14 \times 14 \times 120$ | bneck 7x7 | 160 | 608 | 2 |
| $7 \times 7 \times 160$ | bneck 7x7 | 160 | 576 | 1 |
| $7 \times 7 \times 160$ | bneck 5x5 | 160 | 608 | 1 |
| $7 \times 7 \times 160$ | bneck 3x3 | 160 | 448 | 1 |
| $7 \times 7 \times 160$ | bneck 3x3 | 280 | 1216 | 1 |
| $7 \times 7 \times 280$ | pool, 7x7 | | | 1 |
| $7 \times 7 \times 280$ | conv2d 1x1 | 1920 | | 1 |
| $7 \times 7 \times 1920$ | conv2d 1x1 | 1001 | | 1 |

[11-12M]: [CDD(11M, 0.45)]

3rd layer: Choose ACD and CDD
ACD generates ACA (5M drop), ACB(6M drop), ACC(7M drop), all children searched
CDD generates CDC(10M drop), CDB(9M, 0.4), CDA(8M, 0.37), all children searched
[8-9M]: [CAD(8M, 0.4), ADD(9M, 0.3) CDB(9M, 0.4)]

Start from 1st layer again
1st layer: train and eval BDD (10M, 0.35), DDD(12M, 0.5)
ADD (9M, 0.3)[x], BDD (10M, 0.35), CDD (11M, 0.45), DDD(12M, 0.5)

2nd layer: Choose BDD, CDD
BDD generates BAD (7M drop), BCD (9M, 0.33), BBD (8M, 0.32), all children searched
CDD generates CCD(10M, drop), all children searched
[6-7M]: []
[8M]: [ACD(8M, 0.27) (BBD is better, remove ACD), CAD(8M, 0.4), BBD(8M, 0.32)]
[9-10M]: [ADD(9M, 0.3), BCD(9M, 0.33)]
[11-12M]: [CDD(11M, 0.45)[x]]

3rd layer: Choose BCD, BBD
BCD: BCA(6M, drop), BCB(7M, drop), BCC(8M, 0.32), all children searched
BBD: BBA(5M, drop), BBC(7M, drop), BBB(6M, drop), all children searched
[8-9M]: [CAD(8M, 0.4), CDB(9M, 0.4)]

Move to 1st layer
1st layer:

Table 13: 600M MAdds Search Space

| Operator | # Output filter | # Expanded Filter | strides | \|𝕊\| |
|---|---|---|---|---|
| Conv2d{3x3} | 32 | | 2 | |
| bneck {3x3} | {36, 32, 28, 24, 20, 16} | | 1 | 6 |
| Block filter | {56, 52, 48, 44, 40, 36, 32, 28} | | | 8 |
| bneck {3x3, 5x5} | | {88, 80, 72, 64, 56, 52, 48} | 2 | 14 |
| bneck {3x3, 5x5} | | {88, 80, 72, 64, 56, 52, 48} | 1 | 14 |
| bneck {3x3, 5x5} | | {88, 80, 72, 64, 56, 52, 48} | 1 | 14 |
| bneck {3x3, 5x5} | | {88, 80, 72, 64, 56, 52, 48} | 1 | 14 |
| Block filter | {72, 64, 60, 56, 52, 48, 44, 40} | | | 8 |
| bneck {3x3, 5x5} | | {180, 160, 144, 128, 120, 112, 104, 96, 92, 88, 80} | 2 | 22 |
| bneck {3x3, 5x5, 7x7} | | {240, 220, 200, 180, 160, 140, 130, 120, 100} | 1 | 27 |
| bneck {3x3, 5x5, 7x7} | | {240, 220, 200, 180, 160, 140, 130, 120, 100} | 1 | 27 |
| bneck {3x3, 5x5, 7x7} | | {240, 220, 200, 180, 160, 140, 130, 120, 100} | 1 | 27 |
| bneck {3x3, 5x5, 7x7} | | {240, 220, 200, 180, 160, 140, 130, 120, 100} | 1 | 27 |
| Block filter | {200, 180, 160, 140, 120, 100, 90, 80} | | | 8 |
| bneck {3x3, 5x5, 7x7} | | {440, 400, 360, 320, 280, 260, 240, 200} | 2 | 24 |
| bneck {3x3, 5x5, 7x7} | | {560, 520, 480, 440, 400, 360, 320, 280, 240} | 1 | 27 |
| bneck {3x3, 5x5, 7x7} | | {560, 520, 480, 440, 400, 360, 320, 280, 240} | 1 | 27 |
| bneck {3x3, 5x5, 7x7} | | {560, 520, 480, 440, 400, 360, 320, 280, 240} | 1 | 27 |
| Block filter | {180, 160, 144, 128, 120, 104, 96, 88, 80} | | | 9 |
| bneck {3x3, 5x5, 7x7} | | {560, 520, 480, 440, 400, 360, 320, 280, 240} | 1 | 27 |
| bneck {3x3, 5x5, 7x7} | | {560, 528, 496, 464, 432, 400, 368, 336, 304, 288, 272, 256} | 1 | 36 |
| bneck {3x3, 5x5, 7x7} | | {560, 528, 496, 464, 432, 400, 368, 336, 304, 288, 272, 256} | 1 | 36 |
| bneck {3x3, 5x5, 7x7} | | {560, 528, 496, 464, 432, 400, 368, 336, 304, 288, 272, 256} | 1 | 36 |
| bneck {3x3, 5x5, 7x7} | | {560, 528, 496, 464, 432, 400, 368, 336, 304, 288, 272, 256} | 1 | 36 |
| Block filter | {320, 288, 256, 224, 192, 160} | | | 6 |
| bneck {3x3, 5x5, 7x7} | | {992, 928, 864, 800, 736, 672, 608, 576, 512} | 2 | 27 |
| bneck {3x3, 5x5, 7x7} | | {992, 928, 864, 800, 736, 672, 608, 576, 512} | 1 | 27 |
| bneck {3x3, 5x5, 7x7} | | {992, 928, 864, 800, 736, 672, 608, 576, 512} | 1 | 27 |
| bneck {3x3, 5x5, 7x7} | | {992, 928, 864, 800, 736, 672, 608, 576, 512} | 1 | 27 |
| bneck {3x3, 5x5, 7x7} | | {992, 928, 864, 800, 736, 672, 608, 576, 512} | 1 | 27 |
| bneck {3x3, 5x5, 7x7} | {600, 560, 520, 480, 440, 400, 360, 320} | {1920, 1856, 1792, 1728, 1664, 1600, 1536, 1440} | 1 | 24 |
| pool, 7x7 conv2d 1x1 | {2560, 2240, 1920} | | | 3 |
| conv2d 1x1 | {1001} | | | |

Table 14: LayerNAS Model under 600M MAdds

| Input | Operator | # Output filter | # Expanded Filter | strides |
|---|---|---|---|---|
| $224 \times 224 \times 3$ | conv2d 3x3 | 32 | | 2 |
| $112 \times 112 \times 32$ | bneck 3x3 | 36 | | 1 |
| $112 \times 112 \times 36$ | bneck 5x5 | 36 | 80 | 2 |
| $56 \times 56 \times 36$ | bneck 5x5 | 36 | 72 | 1 |
| $56 \times 56 \times 36$ | bneck 3x3 | 36 | 80 | 1 |
| $56 \times 56 \times 36$ | bneck 5x5 | 36 | 72 | 1 |
| $56 \times 56 \times 36$ | bneck 3x3 | 48 | 144 | 2 |
| $28 \times 28 \times 48$ | bneck 3x3 | 48 | 140 | 1 |
| $28 \times 28 \times 48$ | bneck 3x3 | 48 | 160 | 1 |
| $28 \times 28 \times 48$ | bneck 3x3 | 48 | 130 | 1 |
| $28 \times 28 \times 48$ | bneck 5x5 | 48 | 140 | 1 |
| $28 \times 28 \times 48$ | bneck 7x7 | 140 | 360 | 2 |
| $14 \times 14 \times 140$ | bneck 5x5 | 140 | 360 | 1 |
| $14 \times 14 \times 140$ | bneck 3x3 | 140 | 560 | 1 |
| $14 \times 14 \times 140$ | bneck 5x5 | 140 | 440 | 1 |
| $14 \times 14 \times 140$ | bneck 7x7 | 144 | 360 | 1 |
| $14 \times 14 \times 144$ | bneck 5x5 | 144 | 560 | 1 |
| $14 \times 14 \times 144$ | bneck 3x3 | 144 | 288 | 1 |
| $14 \times 14 \times 144$ | bneck 5x5 | 144 | 400 | 1 |
| $14 \times 14 \times 144$ | bneck 5x5 | 144 | 256 | 1 |
| $14 \times 14 \times 144$ | bneck 3x3 | 192 | 864 | 2 |
| $7 \times 7 \times 192$ | bneck 5x5 | 192 | 928 | 1 |
| $7 \times 7 \times 192$ | bneck 7x7 | 192 | 736 | 1 |
| $7 \times 7 \times 192$ | bneck 7x7 | 192 | 800 | 1 |
| $7 \times 7 \times 192$ | bneck 3x3 | 192 | 928 | 1 |
| $7 \times 7 \times 192$ | bneck 3x3 | 320 | 1440 | 1 |
| $7 \times 7 \times 320$ | pool, 7x7 | | | 1 |
| $7 \times 7 \times 320$ | conv2d 1x1 | 2560 | | 1 |
| $7 \times 7 \times 2560$ | conv2d 1x1 | 1001 | | 1 |

ADD (9M, 0.3)[x], BDD (10M, 0.35)[x], CDD (11M, 0.45)[x], DDD(12M, 0.5)

2nd layer: Choose DDD
DDD: DDA(9M, 0.37), DDB(10M, drop), DDC(11M, drop), all children searched
[6-7M]: []
[8M]: [CAD(8M, 0.4), BBD(8M, 0.32)[x]]
[9-10M]: [ADD(9M, 0.3), BCD(9M, 0.33)[x], DDA(9M, 0.37)]
[11-12M]: [CDD(11M, 0.45)[x]]

3rd layer: Choose CAD, DDA
CAD: CAA(5M, drop), CAC(7M, drop), CAB(6M, drop), all children searched
DDA: DDB(10M, drop), DDC(11M, drop), all children searched

No more potential candidates, the best model found is CAD(8M, 0.4). Out of a total of $4^3 = 64$ candidates, LayerNAS train and eval 12 candidates.

