# OpenReview forum: "LayerNAS: Neural Architecture Search in Polynomial Complexity"
_ICLR.cc/2024/Conference — Submitted to ICLR 2024_

### Official Review · Reviewer_4co7 · 2023-10-31

**Soundness:** 2 fair
**Presentation:** 2 fair
**Contribution:** 2 fair
**Rating:** 5
**Confidence:** 5

**Summary:**

The author attempted to solve NAS via dynamic programming. In order to do so, they made an approximation about the search space that the optimal decision for the i-th layer does not depend on the decision for layers afterward, i.e., the searching problem is simplified to satisfy the optimal substructure requirement (e.g., an optimal solution can be obtained from optimal solutions of its subproblems). In order to make the search complexity manageable, the proposed method, LayerNAS, relies on a grouping/bucketing function which splits the search space into groups/buckets, with each group/bucket only keeping a small amounts of model architectures.

**Strengths:**

The authors provided a new way of tackling the search problem in NAS: dividing the search space into sub-problems and adding the assumption that satisfy the requirements of dynamic programming. To the best of my knowledge, no one has done similar things before. In the task of size search on ImageNet, and both size search and topology search on NATS-Bench, the authors demonstrated the effectiveness of LayerNAS.

In general, the idea is clearly described and easy to follow.

**Weaknesses:**

- The whole idea of LayerNAS is based on the assumption that the optimal decision for the i-th layer does not depend on the decision for the succeeding layers. The paper didn't investigate the soundness of this assumption. For example, in algorithm 1, for a certain layer l and a certain value of h, only a few models with better performance are kept. Is it possible that in the final optimal model, the selected options for some layer i are different from ones that are kept during the search? This situation may become more likely given that each candidate is only trained for a small amount of epochs (e.g., 5 epochs used in the paper), as some architectures are easier to converge (e.g., showing lower loss at the early stage) but cannot keep the momentum till the end (e.g., the loss stop decreasing and the model is eventually surpassed by models with higher loss at the early stage).

- In Table 2, the comparisons stops at FLOPs 627M. How does LayerNAS compare with EfficientNet-B1 and B2? It seems that the comparison with OFA is missing. OFA achieves 76.9% at 230M FLOPs and 80.0% at ~600M FLOPs.

- It is unclear to me how LayerNAS can save some search cost by designing the mapping function $\varphi$, in the case of topology search. The search space in the experiment "NATS-Bench topology search" is too small.

**Questions:**

I'd like to see the authors' response to my questions in the weakness section, especially the second and the third question. For the first question, it is acceptable that LayerNAS may miss some promising architectures under the assumption as long as the final model has good performance.

---

> ### Author Response · Authors · 2023-11-16
>
> ## Weakness 1
> > "...The paper didn't investigate the soundness of this assumption...only a few models with better performance are kept...the model is eventually surpassed by models with higher loss at the early stage"
>
> **Soundness of Assumption 4.1**:
>
> Assumption 4.1 is a common practice in many NAS methods, often employed implicitly rather than explicitly stated. Our phrasing may have inadvertently conveyed the impression that restricting the number of candidates per layer by greedy selection. However, by assigning a unique identifier to each model architecture, LayerNAS retains all possible candidates throughout the search process. A detailed explanation of the **search space completeness** is provided in Appendix E:
>
> > Assume we are searching for the optimal model $s_1..s_n$, and we store all possible model candidates on each layer. During the search process on $layer_n$, we generate model architectures by modifying $o_n$ to other options in $S$. Since we store all model architectures for $layer_{n-1}$, the search process can create all $|S|^n$ candidates on $layer_n$ by adding each $s_n \in S$ to the models in $M_{n-1}$. Therefore, $M_n$ contains all possibilities in the search space. This process can then be applied backward to the first layer.
>
> **LayerNAS methodology**:
>
> LayerNAS framework stands out for its rigorous approach and does not rely on implicit assumptions or search strategy from intuition. LayerNAS is grounded in a set of explicitly defined assumptions that guide its design and operation. This ensures that the search process is sound and well-founded.
>
> Here’s our methodology:
> 1. Explicitly express the underlying assumptions that inform the design of LayerNAS.
> 2. Construct or transform a search space to align with the specified assumptions.
> 3. Rigorously derive LayerNAS from the established assumptions, ensuring that the search strategy is theoretically sound and well-founded.
> 4. Validate the effectiveness of LayerNAS by comparing its performance against established benchmarks, demonstrating its ability to identify high-performance model architectures.
>
> **Correlation between 5-epoch job and full-epoch job**
>
> Thank you for raising this important point. I absolutely concur that training models with fewer epochs tends to favor architectures that converge more readily. This challenge is pervasive across the NAS landscape. Regardless of the model architecture discovered by a NAS method, it is difficult to directly compare its performance to that of a fully trained model. The majority of multi-trial NAS techniques employ a similar approach to LayerNAS, relying on the accuracy of candidates trained with fewer epochs. One-shot NAS employs a different mechanism to train a supernet, but the question remains as to how effectively a candidate selected from the super-net can represent a model trained independently. Training-free NAS approaches utilize significantly faster metrics, which may have lower Kendall rank correlation coefficient.
>
> As previously discussed, LayerNAS does not employ greedy selection per layer. Assumption 4.2 plays a pivotal role in reducing search space complexity by eliminating candidates with similar costs but inferior performance. LayerNAS focuses on size search or compression search problems, modifying only one layer at a time. Model candidates with similar costs that are grouped into the same bucket are unlikely to exhibit significant differences during the size search process.
>
> Moreover, we have provided a detailed explanation in Appendix F, outlining how our approach effectively mitigates this side effect.
>
> > From experiments on MobileNet, we observed that multiple runs on the same model architecture can yield standard deviations of accuracy ranging from 0.08\% to 0.15\%. Often times, the difference can be as high as 0.3\%. To address this, we propose storing multiple candidates for the same cost to increase the likelihood of keeping the better model architecture for every layer search.
>
> > Suppose we have two models with the same cost, $x$ and $y$, where $x$ is inferior and $y$ is superior, and the training accuracy follows a Gaussian distribution $N(\mu, \sigma^2)$. The probability of $x$ obtaining a higher accuracy than $y$ is $P(x - y > 0)$, where $x - y \sim N(\mu_x - \mu_y, {\sigma_x}^2 + {\sigma_y}^2)$. In emprical examples, $\mu_x - \mu_y = -0.002$ and $\sigma_x = 0.001$, then $x$ has the probability of 9.2\% of obtaining a higher accuracy. When we have $L = 20$ layers, the probability of keeping the better model architecture for every layer search is $(1-p)^{20}=18\\%$.
>
> > By storing $k$ candidates with the same cost, we can increase the probability of keeping the better model architecture. When $k=3$, the probability of storing all inferior models is $p^k=0.08\%$. The probability of keeping the better model architecture for all $L=20$ layer searches is 98.4\%, which is practically good enough.

---

> ### Author Response · Authors · 2023-11-16
>
> ## Weakness 2
> > "...How does LayerNAS compare with EfficientNet-B1 and B2? ... comparison with OFA is missing..."
>
> While we have not conducted experiments on EfficientNet architectures or models exceeding 1B MAdds, we are confident that LayerNAS would exhibit superior performance when applied to EfficientNet. This confidence stems from the fact that EfficientNet employs MNasNet search methods, which are superseded by more effective LayerNAS.
>
> OFA and many other works utilize distillation to enhance accuracy. Without ablation study, we believe that this technique introduces additional factors when evaluating the true performance of the model architecture itself. In our experiments, we observed that a simple self-distillation can boost ImageNet top-1 accuracy by 1.5% to 2.5%. This highlights the potential for distillation to artificially inflate performance metrics.
>
> To provide a more objective and reliable comparison, we have included results from papers that conducted ablation studies, where the impact of distillation is clearly isolated. These comparisons demonstrate that LayerNAS outperforms other methods when distillation is not employed. We are open to comparisons with more works, provided that they include ablation studies that isolate the impact of distillation and hyper-parameter tuning.
>
> |Model|Top-1 Acc|MAdds|
> |-|-|-|
> |MobileNetV3-L|75.2|219M|
> |BigNAS-S w/o distillation|75.3|242M|
> |LayerNAS w/o distillation|75.6|229M|
> |BigNAS-S with distillation|76.5|242M|
> |OFA w/ PS #75 w/distillation|76.9|230M|
> |BigNAS-L w/o distillation|78.2|586M|
> |LayerNAS w/o distillation|78.6|627M|
> |BigNAS-L with distillation|79.5|586M|
> |OFALarge w/ PS #75 w/distillation|80.0|595M|
>
> In LayerNAS experiments, we meticulously reproduced the exact same results of MobileNetV3 reported in their paper, establishing a fair baseline for evaluation. For all experiments of LayerNAS on ImageNet, we use the same training setting and do not tune hyper parameters, such as learning rate. This deliberate decision to exclude techniques like distillation, auto-augmentation, and hyperparameter tuning, which could potentially enhance performance, was made to ensure a clear and unbiased assessment of the effectiveness of our proposed method.  By maintaining strict experimental controls, we can confidently attribute any performance gains to the LayerNAS method itself, rather than extrinsic factors. This rigorous approach allows us to isolate the true impact of our method and provides a solid foundation for future comparisons.
>
> ## Weakness 3
> > "...how LayerNAS can save some search cost by designing the mapping function $\phi$ in the case of topology search..."
>
> In Section 4.2, we outlined the primary objective of LayerNAS as developing an efficient algorithm for size search or compression search problems. However, to validate the effectiveness of Assumption 4.1 and demonstrate the generalizability of LayerNAS, we conducted experiments on topology search. In our implementation for topology search, we assigned a unique identifier $\varphi$ to each model, which does not limit the search complexity.
>
> Exploring techniques to constrain search complexity in topology search problems is an avenue for further investigation. Incorporating encoding methods for graph semi-isomorphism could prove effective in grouping similar architectures and optimizing the search process. However, this aspect is beyond the scope of this paper.

---

### Official Review · Reviewer_sD5g · 2023-10-31

**Soundness:** 2 fair
**Presentation:** 2 fair
**Contribution:** 2 fair
**Rating:** 5
**Confidence:** 3

**Summary:**

This papers shows layerwise NAS approach to search a neural architecture layer by layer under computational constraints.

**Strengths:**

1. They propose a new layerwise NAS approach for search neural architecture under constraints.
2. LayerNAS can find out some interesting architectures that outperform the previous NAS algorithms.

**Weaknesses:**

1. What does the improvement of LayerNAS networks over other networks actually come from is uncertain. As mentioned in the end of Sec. 5.1, the architecture mechanisms of the searched networks in this work and other networks are not the same: for example, the authors used SE and Swish while others did not. These details including the undisclosed training strategy (like the learning, weight decay, data augmentation) might largely affect the accuracy, as demonstrated in [1], ConvNeXt [2], and many other followups. This is the key issues to evaluate this paper. Without the claim of using the exact same architecture and training strategy for fair comparison across methods, it is hard to evaluate this paper.

2. Strong assumption based. This paper has to search per-layer. For layer i, it has to assume all the succeeding layers use the default operation (e.g. the most expensive operation) as stated in the first paragraph of Sec. 4. There is no theoretical analysis why this simple and strong assumption leads to better searched architecture than other NAS algorithms.

3. Lack of literature.  Named as LayerNAS, this work lacks comparisons to pioneering work in layerwise NAS: SGAS [3], TNAS [4], and many other followups. Please compare with these works in related work.

[1] Steiner, Andreas, Alexander Kolesnikov, Xiaohua Zhai, Ross Wightman, Jakob Uszkoreit, and Lucas Beyer. "How to train your vit? data, augmentation, and regularization in vision transformers." arXiv preprint arXiv:2106.10270 (2021).
[2] Liu, Zhuang, Hanzi Mao, Chao-Yuan Wu, Christoph Feichtenhofer, Trevor Darrell, and Saining Xie. "A convnet for the 2020s." In Proceedings of the IEEE/CVF conference on computer vision and pattern recognition, pp. 11976-11986. 2022.
[3] Li, Guohao, Guocheng Qian, Itzel C. Delgadillo, Matthias Muller, Ali Thabet, and Bernard Ghanem. "Sgas: Sequential greedy architecture search." In Proceedings of the IEEE/CVF Conference on Computer Vision and Pattern Recognition, pp. 1620-1630. 2020.
[4] Qian, Guocheng, Xuanyang Zhang, Guohao Li, Chen Zhao, Yukang Chen, Xiangyu Zhang, Bernard Ghanem, and Jian Sun. "When NAS Meets Trees: An Efficient Algorithm for Neural Architecture Search." In Proceedings of the IEEE/CVF Conference on Computer Vision and Pattern Recognition, pp. 2782-2787. 2022.

**Questions:**

A detailed example of LayerNAS could have been provided. For example, you can show the step-by-step details of LayerNAS on ImageNet. What are the 100 candidates in each searching layer and which one is chosen.

---

> ### Author Response · Authors · 2023-11-16
>
> ## Question
> > "A detailed example of LayerNAS ... What are the 100 candidates ..."
>
> Our experiments generated thousands of model candidates from 20-layer search space. Sharing the entire log would be impractical and overwhelming for readers. Moreover, during our own analysis of this data, we found that it did not yield significant additional insights.   Therefore, to enhance readability and clarity, we have included a step-by-step explanation of the 3-layer search process in the appendix. This simplified example serves as a concise yet informative illustration of LayerNAS.
>
> Consider a model with 3 layers, each with 4 options: A, B, C, D, corresponding to computational costs of 1M, 2M, 3M, and 4M MAdds, respectively. A sample model architecture can be represented as BCA, indicating that the 1st layer uses B, the 2nd layer uses C, and the 3rd layer uses A. The total cost of this model is 2+3+1=6M MAdds. The goal is to search for the optimal model architecture within the cost range of 8-9M MAdds.
>
> LayerNAS settings:
> * For each layer, candidates are grouped into 4 buckets, each bucket stores up to 2 candidates.
> * In each iteration, 2 candidates are randomly selected to generate 2 valid children.
>
> Cost range:
> * 1st layer cost range: 9-12M, buckets: [9M], [10M], [11M], [12M]
> * 2nd layer cost range: 6-12M, buckets: [6-7M], [8M], [9M, 10M], [11M, 12M]
> * 3rd layer: only stores [8M, 9M]
>
> Marks:
> * Candidates that fall outside the designated cost range are marked as "drop"
> * Once all child architectures have been generated, the model is marked with "[x]"
>
> ```
> 1st layer: train and eval: ADD (9M, 0.3), CDD (11M, 0.45)
> 2nd layer: Choose ADD and CDD
> 	ADD generates ABD (7M drop), ACD (8M, 0.27), AAD(6M drop), all children searched
> 	CDD generates CAD(8M, 0.4), CBD(9M, 0.42)
> 	[6-7M]: []
> 	[8M]: [ACD(8M, 0.27), CAD(8M, 0.4)]
> 	[9-10M]: [ADD(9M, 0.3)]
> 	[11-12M]: [CDD(11M, 0.45)]
> 3rd layer: Choose ACD and CDD
> 	ACD generates ACA (5M drop), ACB(6M drop), ACC(7M drop), all children searched
> 	CDD generates CDC(10M drop), CDB(9M, 0.4), CDA(8M, 0.37), all children searched
> 	[8-9M]: [CAD(8M, 0.4), ADD(9M, 0.3) CDB(9M, 0.4)]
>
> Start from 1st layer again
> 1st layer: train and eval BDD (10M, 0.35), DDD(12M, 0.5)
> 	ADD (9M, 0.3)[x], BDD (10M, 0.35), CDD (11M, 0.45), DDD(12M, 0.5)
> 2nd layer: Choose BDD, CDD
> 	BDD generates BAD (7M drop), BCD (9M, 0.33), BBD (8M, 0.32), all children searched
> 	CDD generates CCD(10M, drop), all children searched
> 	[6-7M]: []
> 	[8M]: [ACD(8M, 0.27) (BBD is better, remove ACD), CAD(8M, 0.4), BBD(8M, 0.32)]
> 	[9-10M]: [ADD(9M, 0.3), BCD(9M, 0.33)]
> 	[11-12M]: [CDD(11M, 0.45)[x]]
> 3rd layer: Choose BCD, BBD
> 	BCD: BCA(6M, drop), BCB(7M, drop), BCC(8M, 0.32), all children searched
> 	BBD: BBA(5M, drop), BBC(7M, drop), BBB(6M, drop), all children searched
> 	[8-9M]: [CAD(8M, 0.4), CDB(9M, 0.4)]:
>
> Move to 1st layer
> 1st layer:
> 	ADD (9M, 0.3)[x], BDD (10M, 0.35)[x], CDD (11M, 0.45)[x], DDD(12M, 0.5)
> 2nd layer: Choose DDD
> 	DDD: DDA(9M, 0.37), DDB(10M, drop), DDC(11M, drop), all children searched
> 	[6-7M]: []
> 	[8M]: [ CAD(8M, 0.4), BBD(8M, 0.32)[x]]
> 	[9-10M]: [ADD(9M, 0.3), BCD(9M, 0.33)[x], DDA(9M, 0.37)]
> 	[11-12M]: [CDD(11M, 0.45)[x]]
> 3rd layer: Choose CAD, DDA
> 	CAD: CAA(5M, drop),  CAC(7M, drop), CAB(6M, drop), all children searhced
> 	DDA: DDB(10M, drop), DDC(11M, drop), all children searched
>
> No more potential candidates, the best model found is CAD(8M, 0.4)
> Out of a total of 4^3 = 64 candidates, LayerNAS train and eval 12 candidates.
> ```
>
> ## Weakness 1
> > "... the authors used SE and Swish while others did not ... fair comparison across methods..."
>
> Thank you for raising this important point regarding the comparison with benchmark models. We have taken great care to ensure that our comparisons are fair and transparent. We recognize that many NAS works incorporate SE+Swish, and many others don't. In Table 5.1, we explicitly present the results for both model architectures discovered by LayerNAS, with and without SE+Swish, and mark them in different symbols. This allows readers to make fair comparisons with other models under different configurations.
>
>
> We meticulously reproduced the exact same result of MobileNetV3 reported in their paper, establishing a fair baseline for evaluation. For all experiments of LayerNAS on ImageNet, we use the same training setting and do not finetune hyper parameters, such as learning rate.
> Distillation and auto-augmentation could further improve the performance of our models, a simple self-distillation can improve accuracy by 1.5% to 2.5%. However, we deliberately chose to exclude them in our experiments, because their inclusion would make it difficult to determine the true contribution of LayerNAS objectively. We believe that our strict and controlled experimental setup allows for a clear assessment of the effectiveness of our proposed method. Otherwise, as you mentioned, it's hard to evaluate if the improvement comes from distillation or from NAS methods.

---

> > ### Author Response · Authors · 2023-11-16
> >
> > ## Weakness 2
> > > "Strong assumption based... no theoretical analysis..."
> >
> > Assumption 4.1 is a common practice in many NAS methods, often employed implicitly rather than explicitly stated.  For example, SGAS selects an edge greedily and prunes unchosen weights. This sequential selection assumes the following iterations will not be affected by preceding iterations. Our phrasing may have inadvertently conveyed the impression that restricting the number of candidates per layer by greedy selection. However, by assigning a unique identifier to each model architecture, LayerNAS retains all possible candidates throughout the search process. A detailed explanation of the **search space completeness** is provided in Appendix E:
> >
> > > Assume we are searching for the optimal model $s_1..s_n$, and we store all possible model candidates on each layer. During the search process on $layer_n$, we generate model architectures by modifying $o_n$ to other options in $S$. Since we store all model architectures for $layer_{n-1}$, the search process can create all $|S|^n$ candidates on $layer_n$ by adding each $s_n \in S$ to the models in $M_{n-1}$. Therefore, $M_n$ contains all possibilities in the search space. This process can then be applied backward to the first layer.
> >
> > LayerNAS framework stands out for its rigorous approach and does not rely on implicit assumptions or search strategy from intuition. LayerNAS is grounded in a set of explicitly defined assumptions that guide its design and operation. This ensures that the search process is sound and well-founded.
> >
> > Here’s our methodology:
> > 1. Explicitly express the underlying assumptions that inform the design of LayerNAS.
> > 2. Construct or transform a search space to align with the specified assumptions.
> > 3. Rigorously derive LayerNAS from the established assumptions, ensuring that the search strategy is theoretically sound and well-founded.
> > 4. Validate the effectiveness of LayerNAS by comparing its performance against established benchmarks, demonstrating its ability to identify high-performance model architectures.
> >
> > LayerNAS offers a valuable contribution to the field of NAS research by demonstrating that a simple yet well-founded method can achieve remarkable results, even under stringent and unbiased experimental conditions. This finding challenges the prevailing notion that NAS methods require increasing complexity to achieve optimal performance.
> >
> > LayerNAS's simplicity stems from its adherence to a set of clearly defined assumptions, which rigorously derives the search process to ensure its effectiveness. By focusing on essential principles rather than introducing intricate mechanisms, LayerNAS achieves both simplicity and effectiveness.
> >
> >
> > ## Weakness 3
> > > "Lack of literature..."
> >
> > Thank you for bringing these relevant studies to our attention. We have added them to our related works section:
> >
> > > Several prior works, such as \cite{liu2018progressive}, \cite{li2020sgas}, \cite{qian2022tnas}, have introduced progressive search mechanisms on layerwise search spaces. LayerNAS stands apart by explicitly articulating the underlying assumptions of the layerwise search space, rigorously deriving the method from these assumptions, and effectively constraining the polynomial search space complexity.

---

### Official Review · Reviewer_xmki · 2023-11-05

**Soundness:** 4 excellent
**Presentation:** 3 good
**Contribution:** 3 good
**Rating:** 8
**Confidence:** 4

**Summary:**

This paper propose a simple method to break down the neural architecture search approach into a layer-wise one. Specifically, for a search space of L-layer network, it only searchs for one layer at each training iteration instead of all layers. Experiments are conducted on MobileNetV2, MobileNet-V3, NASBench101 and NATS-Bench spaces.

**Strengths:**

The paper is well-written and easy to follow. The authors provide clear explanations and examples throughout the paper.

Breaking down the search problem into a Combinatorial Optimization problem seems novel and interesting, and reducing the search cost to polynomial time, which is clearly a breakthrough to the research community.

LayerNAS can be applied to operation, topology and multi-objective NAS search

Results on ImageNet seems to surpass state-of-the-art methods by a clear margin, evidencing their effectiveness of LayerNAS.

**Weaknesses:**

I do not particular have a question, this paper seems to be easy enough to follow.

**Questions:**

N/A

---

> ### Author Response · Authors · 2023-11-16
>
> Thank you for recognizing our contribution. We really appreciate it.

---

### Official Review · Reviewer_nAfG · 2023-11-08

**Soundness:** 2 fair
**Presentation:** 2 fair
**Contribution:** 2 fair
**Rating:** 5
**Confidence:** 4

**Summary:**

This paper proposes a novel progressive search method and decouples the search constraints from the optimization objective in order to reduce the search space.

**Strengths:**

1. The proposed method provides a new idea for progressive architecture search strategies.
2. Extensive empirical experiments demonstrate the effectiveness of LayerNAS.

**Weaknesses:**

1. Figure 2 is somewhat confusing and seems to have little relevance to the description of Algorithm 1. It would help the reader to understand the details of the algorithm if the authors could give a concrete example of a LayerNAS that contains specific hyperparameters
2. Assumption 4.1 is too strong. This strategy means that a large number of candidate architectures will be ignored. It is promising in terms of experimental performance. However, the authors do not give some theoretical or other analysis to justify their hypothesis.
3. I'm not sure if most of the search space meets the assumptions of the proposed approach. If not, a specific transformation of the search space is necessary to satisfy the assumptions of the search algorithm, however the transformation may be very complex. This makes me concerned about the ease of use and generalizability of the algorithm.

**Questions:**

In the experimental part, the analysis for the hyperparameter $H$ is missing. I am curious how the performance changes when $H$ is greater than 100.

---

> ### Author Response · Authors · 2023-11-16
>
> ## Question
> > In the experimental part, the analysis for the hyperparameter $H$ is missing. I am curious how the performance changes when $H$ is greater than 100
>
> We did not explore larger values of $H$ because of the following considerations:
> * **Marginal improvement**: The performance gain from increasing $H$ beyond a certain point may be marginal, potentially masked by variations in different training runs. Even if the optimal model architecture does lie within this larger search space, the marginal improvement in accuracy might not be substantial enough to offset the increased search cost.
> * **Computational cost**: Doubling $H$ doubles the computational cost of the search process. Given the limited computational resources available, we opted to focus on smaller values of $H$ to achieve a balance between performance and efficiency.
>
> Despite these limitations, we conducted experiments with $H=20$ under a 300M MAdds constraint. The best model architecture obtained without SE modules achieved a top-1 accuracy of 74.6% on ImageNet. Increasing $H$ to 100 further improved the accuracy to 75.5%, demonstrating the potential benefits of larger $H$ values.
>
> We anticipate that plotting model accuracy against $H$ would yield a curve with a steep initial ascent followed by a gradual flattening. This pattern reflects the diminishing returns of increasing $H$ beyond a certain point. Initially, larger values of $H$ provide more opportunities to explore the search space, leading to significant improvements in accuracy. However, as $H$ increases, the potential for further gains diminishes, and the curve approaches an asymptotic limit. Theoretically, if we set unlimited $H$ and unlimited duplicates per bucket, the search space becomes exhaustively explored, akin to a brute-force search, guaranteeing an optimal result.
>
> ## Weakness 1
> > "...give a concrete example of a LayerNAS..."
>
> Thanks for pointing it out. We have updated the paper with the new figure, and prepared a step-by-step example to illustrate the search process on a simplified 3-layer search space in the appendix.
>
> Consider a model with 3 layers, each with 4 options: A, B, C, D, corresponding to computational costs of 1M, 2M, 3M, and 4M MAdds, respectively. A sample model architecture can be represented as BCA, indicating that the 1st layer uses B, the 2nd layer uses C, and the 3rd layer uses A. The total cost of this model is 2+3+1=6M MAdds. The goal is to search for the optimal model architecture within the cost range of 8-9M MAdds.
>
> LayerNAS settings:
> * For each layer, candidates are grouped into 4 buckets, each bucket stores up to 2 candidates.
> * In each iteration, 2 candidates are randomly selected to generate 2 valid children.
>
> Cost range:
> * 1st layer cost range: 9-12M, buckets: [9M], [10M], [11M], [12M]
> * 2nd layer cost range: 6-12M, buckets: [6-7M], [8M], [9M, 10M], [11M, 12M]
> * 3rd layer: only stores [8M, 9M]
>
> Marks:
> * Candidates that fall outside the designated cost range are marked as "drop"
> * Once all child architectures have been generated, the model is marked with "[x]"
>
> ```
> 1st layer: train and eval: ADD (9M, 0.3), CDD (11M, 0.45)
> 2nd layer: Choose ADD and CDD
> 	ADD generates ABD (7M drop), ACD (8M, 0.27), AAD(6M drop), all children searched
> 	CDD generates CAD(8M, 0.4), CBD(9M, 0.42)
> 	[6-7M]: []
> 	[8M]: [ACD(8M, 0.27), CAD(8M, 0.4)]
> 	[9-10M]: [ADD(9M, 0.3)]
> 	[11-12M]: [CDD(11M, 0.45)]
> 3rd layer: Choose ACD and CDD
> 	ACD generates ACA (5M drop), ACB(6M drop), ACC(7M drop), all children searched
> 	CDD generates CDC(10M drop), CDB(9M, 0.4), CDA(8M, 0.37), all children searched
> 	[8-9M]: [CAD(8M, 0.4), ADD(9M, 0.3) CDB(9M, 0.4)]
>
> Start from 1st layer again
> 1st layer: train and eval BDD (10M, 0.35), DDD(12M, 0.5)
> 	ADD (9M, 0.3)[x], BDD (10M, 0.35), CDD (11M, 0.45), DDD(12M, 0.5)
> 2nd layer: Choose BDD, CDD
> 	BDD generates BAD (7M drop), BCD (9M, 0.33), BBD (8M, 0.32), all children searched
> 	CDD generates CCD(10M, drop), all children searched
> 	[6-7M]: []
> 	[8M]: [ACD(8M, 0.27) (BBD is better, remove ACD), CAD(8M, 0.4), BBD(8M, 0.32)]
> 	[9-10M]: [ADD(9M, 0.3), BCD(9M, 0.33)]
> 	[11-12M]: [CDD(11M, 0.45)[x]]
> 3rd layer: Choose BCD, BBD
> 	BCD: BCA(6M, drop), BCB(7M, drop), BCC(8M, 0.32), all children searched
> 	BBD: BBA(5M, drop), BBC(7M, drop), BBB(6M, drop), all children searched
> 	[8-9M]: [CAD(8M, 0.4), CDB(9M, 0.4)]
>
> Move to 1st layer
> 1st layer:
> 	ADD (9M, 0.3)[x], BDD (10M, 0.35)[x], CDD (11M, 0.45)[x], DDD(12M, 0.5)
> 2nd layer: Choose DDD
> 	DDD: DDA(9M, 0.37), DDB(10M, drop), DDC(11M, drop), all children searched
> 	[6-7M]: []
> 	[8M]: [CAD(8M, 0.4), BBD(8M, 0.32)[x]]
> 	[9-10M]: [ADD(9M, 0.3), BCD(9M, 0.33)[x], DDA(9M, 0.37)]
> 	[11-12M]: [CDD(11M, 0.45)[x]]
> 3rd layer: Choose CAD, DDA
> 	CAD: CAA(5M, drop),  CAC(7M, drop), CAB(6M, drop), all children searhced
> 	DDA: DDB(10M, drop), DDC(11M, drop), all children searched
>
> No more potential candidates, the best model found is CAD(8M, 0.4)
> Out of a total of 4^3 = 64 candidates, LayerNAS train and eval 12 candidates.
> ```

---

> > ### Author Response · Authors · 2023-11-16
> >
> > ## Weakness 2
> > > "Assumption 4.1 is too strong... some theoretical or other analysis to justify their hypothesis."
> >
> > Assumption 4.1 is a common practice in many NAS methods, often employed implicitly rather than explicitly stated. Our phrasing may have inadvertently conveyed the impression that restricting the number of candidates per layer by greedy selection. However, by assigning a unique identifier to each model architecture, LayerNAS retains all possible candidates throughout the search process. A detailed explanation of the **search space completeness** is provided in Appendix E:
> >
> > > Assume we are searching for the optimal model $s_1..s_n$, and we store all possible model candidates on each layer. During the search process on $layer_n$, we generate model architectures by modifying $o_n$ to other options in $S$. Since we store all model architectures for $layer_{n-1}$, the search process can create all $|S|^n$ candidates on $layer_n$ by adding each $s_n \in S$ to the models in $M_{n-1}$. Therefore, $M_n$ contains all possibilities in the search space. This process can then be applied backward to the first layer.
> >
> >
> > LayerNAS framework stands out for its rigorous approach and does not rely on implicit assumptions or intuitive search strategy. LayerNAS is grounded in a set of explicitly defined assumptions that guide its design and operation. This ensures that the search process is sound and well-founded.
> >
> > Here’s our methodology:
> > 1. Explicitly express the underlying assumptions that inform the design of LayerNAS.
> > 2. Construct or transform a search space to align with the specified assumptions.
> > 3. Rigorously derive LayerNAS from the established assumptions, ensuring that the search strategy is theoretically sound and well-founded.
> > 4. Validate the effectiveness of LayerNAS by comparing its performance against established benchmarks, demonstrating its ability to identify high-performance model architectures.
> >
> >
> >
> > ## Weakness 3
> > > "if most of the search space meets the assumptions ... the transformation may be very complex ... the ease of use and generalizability of the algorithm."
> >
> > Most popular models follow a layer-by-layer structure. To avoid ambiguity in our discussion, let's establish clear definitions for two key terms:
> > * architecture layer: a layers that constitutes the model
> > * search layer: a decision point during the search process, determining the one configuration in one or more architecture layers.
> >
> > If there's a search option that is applied to more than one architecture layers, such as the number of channels in a residual block, it should be considered an individual search layer. In Appendix E, we have an explanation why this sequential search process works:
> >
> > > Assume, after LayerNAS sequential search, we get optimal model defined as $a_1..a_i..a_n$. For sake of contradiction, there exists a model $a_1..b_i..a_n$, with superior performance, by applying a change in previous layers. Since the search space is complete, model $a_1..b_io_{i+1}..o_n$ must exist, and has been processed in $M_i$. In the sequential search, model $a_1..b_ia_{i+1}..o_n$ can be created by using $a_{i+1}$ on $\text{layer}_{i+1}$. Repeating this process for all subsequent layers will eventually lead to $a_1..b_i..a_n$, contradicting our assumption that optimal model from sequential search is $a_1..a_i..a_n$. Therefore, we can search sequentially.
> >
> > While heuristic methods may be employed to refine the search space, the effectiveness of LayerNAS on more general scenarios can be further validated through experiments on NASBench-101 and NATSBench topology search.

---

### Author Response · Authors · 2023-11-21

Dear Reviewers,

We sincerely appreciate your valuable time and insightful reviews of our work. We would like to address a potential misunderstanding regarding Assumption 4.1 that may have arisen from our initial explanation.

Assumption 4.1 is a common practice in the field of Neural Architecture Search (NAS), although it is often not explicitly stated in other works. In LayerNAS, we do not employ a greedy selection process for candidate architectures per layer. Instead, we group all possible candidate architectures for each layer. This approach allows us to explore a broader range of architectural configurations and avoid getting stuck in local optima.

We have provided a detailed explanation and analysis of Assumption 4.1 in our comments. We encourage you to engage in the discussion and provide further feedback on this aspect of our work. Your insights are invaluable in helping us refine our approach and improve the overall quality of our research.

Thank you for your continued attention to our work. We look forward to your continued engagement in the discussion.

---

### Meta-Review · Area_Chair_Sfnz · 2023-12-07

**Metareview:**

LayerNAS, a method proposed in the paper, addresses the challenge of multi-objective Neural Architecture Search (NAS) by transforming it into a combinatorial optimization problem, limiting the search complexity and consistently discovering superior models across various search spaces compared to strong baselines.

Reviewers have consistently shown concerns about "Assumption 4.1 is too strong" and "Experiments effectiveness".

Authors have replied to the above concerns during the rebuttal, but problem remains

- Assumption 4.1 is a common practice in many NAS methods, but this does not mean it is correct. Especially, authors need to compare with more recent methods that do not adopt something similar to "Assumption 4.1". If such empirical results are done and results show LayerNAS is still very competitive, then Assumption 4.1 may not be a problem.

- However, it is not the case. In authors' reply to Reviewer 4co7. LayerNAS is worse than OFA and BigNAS. These two baselines do not adopt a layer-wise approach.

**Justification For Why Not Higher Score:**

While reviewers do not respond to authors, it is clear that Assumption is not reasonable and empirical comparison is not sufficient based on results, which are posted by authors during the rebuttal.

**Justification For Why Not Lower Score:**

N/A.

---

### Decision · Program_Chairs · 2024-01-16

Reject